# Proximity extracellular protein-protein interaction analysis of EGFR using AirID-conjugated fragment of antigen binding

Kohdai Yamada[1], Ryouhei Shioya[1], Kohei Nishino[2], Hirotake Furihata [1], Atsushi Hijikata[3], Mika K. Kaneko [4,5], Yukinari Kato [4,5], Tsuyoshi Shirai [6], Hidetaka Kosako [2]✉ & Tatsuya Sawasaki [1]✉

Receptor proteins, such as epidermal growth factor receptor (EGFR), interact with other proteins in the extracellular region of the cell membrane to drive intracellular signalling. Therefore, analysis of extracellular protein-protein interactions (exPPIs) is important for understanding the biological function of receptor proteins. Here, we present an approach using a proximity biotinylation enzyme (AirID) fusion fragment of antigen binding (FabID) to analyse the proximity exPPIs of EGFR. AirID was C-terminally fused to the Fab fragment against EGFR (EGFR-FabID), which could then biotinylate the extracellular region of EGFR in several cell lines. Liquid Chromatography-Mass Spectrometry (LC-MS/MS) analysis indicated that many known EGFR interactors were identified as proximity exPPIs, along with many unknown candidate interactors, using EGFR-FabID. Interestingly, these proximity exPPIs were influenced by treatment with EGF ligand and its specific kinase inhibitor, gefitinib. These results indicate that FabID provides accurate proximity exPPI analysis of target receptor proteins on cell membranes with ligand and drug responses.

Membrane proteins account for more than 30% of the human proteome and play important roles in many cellular functions, such as environmental responses, signal transduction, and cell-cell interactions[1,2]. Additionally, the extracellular regions of many membrane proteins interact with other proteins for these cellular functions[3]. Therefore, analysis of extracellular protein–protein interactions (exPPIs) is a key issue for understanding the biological functions of membrane proteins. General methods to identify proteins interacting with the target protein include the yeast two-hybrid system[4,5], mass spectrometry analysis after immunoprecipitation[6,7], and cell-free protein arrays[8,9]. However, these methods are not suitable for the exPPI analysis of membrane proteins because they have been

designed specifically for the analysis of soluble proteins localised in the cytoplasm and nucleus. In addition, because many extracellular regions make specific specialised structures on the membrane and extracellular environments, exPPI analysis requires an environment similar to that of the cell membrane.

When considering the exPPI of membrane proteins, it is important to note that the environment is vastly different outside and inside the cell membrane. Furthermore, the biological responses of membrane proteins, such as ligand binding and cell-cell interactions, occur in the extracellular region of the cell membrane. However, because the extracellular domains of many membrane proteins function as regulatory and response regions for localisation, transport, and ligand

[1]Division of Cell-Free Life Science, Proteo-Science Center, Ehime University, 3 Bunkyo-cho, Matsuyama, Ehime 790-8577, Japan. [2]Division of Cell Signaling, Fujii Memorial Institute of Medical Sciences, Institute of Advanced Medical Sciences, Tokushima University, Tokushima 770-8503, Japan. [3]Laboratory of Computational Genomics, School of Life Sciences, Tokyo University of Pharmacy and Life Sciences, Hachioji 192-0392, Japan. [4]Department of Antibody Drug Development, Tohoku University Graduate School of Medicine, Sendai 980-8575, Japan. [5]Department of Molecular Pharmacology, Tohoku University Graduate School of Medicine, Sendai 980-8575, Japan. [6]Department of Bioscience, Nagahama Institute of BioScience and Technology, 1266 Tamura, Nagahama 526-0829, Japan. ✉e-mail: kosako@tokushima-u.ac.jp; sawasaki@ehime-u.ac.jp

binding, using the native protein without a detectable peptide tag or insertion protein is the best way to perform exPPI analysis. Under these conditions, some studies have used a highly specific monoclonal antibody (mAb) against the extracellular domains of the target membrane protein[10,11] because unnatural modifications are not required, and they can be used directly for cell analysis.

Proximity-labelling technology has been widely used to identify partner proteins[12–14]. Since proximity labelling detects proteins that are very close together, it possibly obtains more precise information about interacting proteins[15,16]. Currently, several proximity-labelling-based techniques, such as the enzyme-mediated proximity cell labelling system (EXCELL)[17], pupylation-based interaction tagging (PUP-IT)[18], engineered ascorbate peroxidase (APEX)[19,20], and μMap, have been developed for the interactome analysis of membrane proteins[10]. These technologies were conjugated to proximity-labelling probes for a target-specific mAb. However, because these methods did not identify the labelling sites, it is not clear whether the interaction regions between the target membrane protein and interacting proteins were exPPIs.

BioID (proximity-dependent biotin identification)[21–24] has been developed as an enzymatic proximity-labelling technology. Recently, we developed a new enzyme AirID for the BioID method[24,25]. However, currently, BioID has not been conjugated to mAb and is used as a direct tool to analyse exPPIs through genetic insertions[26]. This genetic approach requires the insertion of BioID enzymes between the domains of membrane proteins, which may have a substantial effect on membrane protein function. Therefore, an exPPI analysis using a protein that fuses a BioID enzyme with an antibody capable of recognising a membrane protein, may be an alternative to the current exPPI analysis approach.

Here, we show an AirID fusion antibody technology suitable for exPPI analysis of membrane proteins. We used a specific AirID-fusion antibody for exPPI analysis of epidermal growth factor receptor (EGFR) in cancer cell lines. We produced modified antibodies by fusing AirID to the fragment of antigen binding (Fab) region (FabID) or to the full-length of the antibody (mAbID). We achieved higher biotinylation efficiency with FabID in vitro and in cells than with mAbID. LC-MS/MS analysis of biotinylated peptides indicated that EGFR-FabID (FabID that recognises EGFR) biotinylated the extracellular regions of many membrane proteins, including well-known EGFR interactors. Gefitinib (Iressa), a pharmaceutical compound widely used for the treatment of lung cancer, is a highly specific EGFR inhibitor for tyrosine kinase activity in the intracellular region[27]. Interestingly, these proximity exPPI interactions were influenced by the EGF ligand and gefitinib treatment, suggesting that ligand binding and inhibition of receptor tyrosine kinase (RTK) activity have the potential to change exPPIs. These results indicate that the FabID system is a useful method for exPPI analysis of receptor proteins.

## Results

### Biotinylation of antigen having an epitope AGIA tag by AGIA-mAbID or AGIA-FabID

We recently created a proximity biotinylation enzyme, AirID, for the BioID method, based on an artificial intelligence algorithm[24]. Because AirID provided an analysis of highly specific PPIs in cells[25], we used the AirID enzyme for proximity biotinylation of antigens. To design an AirID-fusion antibody, the AGIA-tag system we developed previously was utilised because the AGIA peptide tag (EEAAGIARP) is specifically recognised by rabbit anti-AGIA mAb[28], and we had the gene sets for the construction and production of its mAb. Proximity biotinylation prefers a short distance between the antibody and antigen because it occurs within a very short range (~10 nm)[13]. To integrate AirID with anti-AGIA mAb, we attempted to use two types of antibodies: (1) AGIA-mAbID: AirID is fused to the C-terminus of the heavy chain in full-length IgG antibody or (2) AGIA-FabID: AirID is fused to one of the

fragments of antigen binding instead of the Fc region (Fig. 1a). We considered the effect of the presence or absence of the fragment crystallisable region (Fc region) on the biotinylation efficiency.

To obtain AGIA-mAbID and AGIA-FabID proteins, they were expressed in Expi293F cells and purified by affinity chromatography using a nickel column (Fig. 1b) because they have a 10x-histidine tag at their C-terminus. Biotinylation of these purified proteins was mainly found in the heavy chain but not in each light chain (Fig. 1c). To check the biotinylation ability to the antigen (Fig. 1d), FLAG-GST fused to the C-terminal region of human DRD1 (FLAG-GST-DRD1 CTD) was used because the anti-AGIA mAb recognised it as the epitope[28]. Immunoblotting with an anti-biotin antibody showed that antigen biotinylation by AGIA-FabID was higher than that by AGIA-mAbID (arrowhead in Fig. 1e). AGIA-FabID also biotinylated other proteins with an AGIA tag (Fig. 1f). Furthermore, to investigate whether AGIA-FabID could biotinylate a protein interacting with an antigen, an interaction between AGIA-tagged p53 and FLAG-GST-fusion MDM2 (FG-MDM2), previously used to check proximity biotinylation as a PPI model[24], was used (Fig. 1g). AGIA-FabID biotinylated both AGIA-p53 and FG-MDM2 (Fig. 1h) but did not biotinylate FLAG-p53 and FG-MDM2. Taken together, these results showed that FabID could biotinylate both antigens and their interacting proteins.

We previously showed that AirID functions in cells[24,25]. Since the pH conditions of the extracellular environment (pH ~ 7.4[29]) differ from those of the intracellular environment, we investigated the biotinylation activity of AGIA-FabID under various pH conditions. These results showed that AGIA-FabID has biotinylation activity between pH 7 and 8 (Supplementary Fig. 1a), suggesting that FabID can be used in this pH range, mimicking extracellular conditions.

### Biotinylation of EGFR by EGFR-FabID or EGFR-mAbID

EGFR is a critical signalling mediator in many epithelial cancers[30]. EGFR consists of extracellular, transmembrane, and intracellular domains, with the C-terminal domain having tyrosine kinase activity. When a ligand such as EGF or TGF-α binds to EGFR, the structure of the extracellular domain of EGFR is altered, and EGFR forms a homodimer with other EGFR or heterodimers with other ERBB family members. This dimerisation activates the intracellular tyrosine kinase domain, which phosphorylates the tyrosine residues in the C-terminal domain and transmits signals downstream[31]. In addition, EGFR interacts with many membrane proteins[32,33]. However, it is often unclear whether these interactions occur in the extracellular or intracellular regions of EGFR. Thus, we used EGFR as a model membrane protein for proximity exPPI analysis. To make a proximity biotinylation probe, we combined AirID and a previously reported anti-human EGFR mouse mAb (clone EMab-134) because it showed high affinity and selectivity for the extracellular epitope (epitope region no. 377–386) of EGFR[34,35]. The EGFR-FabID and EGFR-mAbID were constructed similarly to the anti-AGIA mAb shown in Fig. 1. Both EGFR-FabID and EGFR-mAbID were purified and confirmed to make disulfide bonds on the Fab and IgG (Fig. 2a). Full-length human EGFR was synthesised using a non-reducing wheat cell-free protein production system because its extracellular region (25–645) has S-S bonds[36,37]. Biotinylation assays showed that full-length human EGFR was biotinylated in vitro by EGFR-FabID and EGFR-mAbID (Fig. 2b) but not by negative controls, AGIA-FabID and AGIA-mAbID.

In a method using an AirID-fused antibody, the accessibility of the AirID-antibody is a very important factor for increasing the detection sensitivity. Many proximity biotinylation methods use suspension cells[10] because antibodies can access the entire surface area of suspension cells. Accordingly, Expi293F cells were used in suspension culture conditions and transfected with a plasmid containing the human EGFR gene (EGFR) or pcDNA3.1 empty plasmid (Mock). To validate the biotinylation ability of cells, EGFR-FabID or EGFR-mAbID was added to the serum-free medium and incubated with Expi293F cells transiently

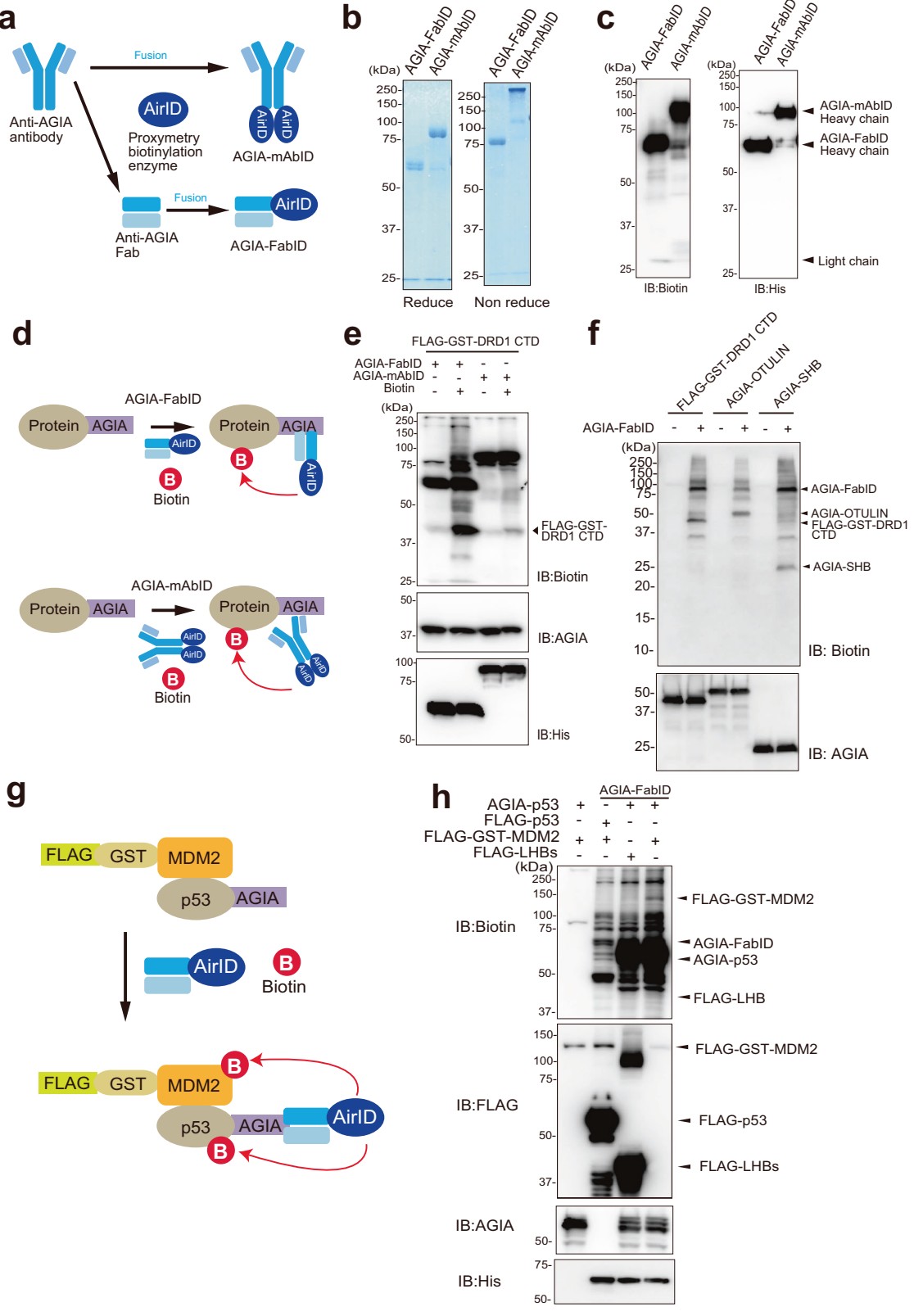

overexpressing EGFR for 2 h (Fig. 2c). Similar to the in vitro experiments above, both EGFR-FabID and EGFR-mAbID clearly biotinylated EGFR, whereas AGIA-FabID did not. A comparison of both probes indicated that the biotinylation efficiency of EGFR-FabID was much higher than that of EGFR-mAbID. The biotinylation of EGFR was also confirmed using a streptavidin-pull-down assay (STA-PDA in Fig. 2c).

To understand why EGFR was more biotinylated in cells treated with EGFR-FabID than with EGFR-mAbID, we performed structural modelling of FabID and mAbID with the extracellular region of EGFR, as previously reported[36]. The modelling showed that the proximity distance between the extracellular domain and EGFR-FabID (Fig. 2d) or EGFR-mAbID (Fig. 2e) is 9.1 and 13.9 nm respectively, indicating that

**Fig. 1 | Biotinylation of AGIA-tag protein by AGIA-FabID and AGIA-mAbID.**
**a** Schematic of AGIA antibody fused with the proximity biotinylation enzyme AirID. AirID is fused to AGIA-Fab or AGIA-mAb, **b** SDS-PAGE and Coomassie Brilliant Blue (CBB) staining in reducing and non-reducing state of AGIA-FabID and AGIA-mAbID synthesised by Expi293F expression system and purified by Ni Sepharose excel. **c** Detection of self-biotinylation of AGIA-FabID and AGIA-mAbID by immunoblotting using an anti-biotin antibody. Detection of AGIA-FabID and AGIA-mAbID using an anti-His tag antibody. **d** Schematic of biotinylation targeting the AGIA-tagged protein using AGIA-FabID and AGIA-mAbID. **e** Comparison of the activities of AGIA-FabID and AGIA-mAbID targeting FLAG-GST-DRD1 CTD (FLAG-GST fused to the C-terminus of DRD1) in vitro. **f** Targeting various AGIA-tagged proteins biotinylated by AGIA-FabID. **g** Schematic of the biotinylation-targeting complex of p53-MDM2 using AGIA-FabID. **h** Confirmation of biotinylation of FLAG-GST-MDM2 and AGIA-p53 by immunoblotting blotting. The SHB *and* LHB are S-protein of hepatitis B virus. **b, c, e, f, h**, The experiment was independently repeated twice, with similar results. Source data are provided as a Source Data file.

EGFR-FabID is less distant than EGFR-mAbID and thus may have greater biotinylation efficiency than EGFR-mAbID.

TurboID, another highly active biotinylation enzyme[22], was also fused with the same Fab against EGFR. The expression of TurboID-fusion Fab was lower than AirID-based EGFR-FabID in Expi293F cells (IB: His in Supplementary Fig. 1b), whereas high biotinylated proteins were found in the culture medium expressing TurboID-based EGFR-FabID (IB: Biotin). Purified TurboID-based EGFR-FabID also showed higher biotinylation than AirID-based EGFR-FabID in the heavy chain fragment (Supplementary Fig. 1c). Moreover, in vitro biotinylation activity of TurboID-based EGFR-FabID was much lower than that of AirID-based EGFR-FabID (arrowhead in Supplementary Fig. 1d). Although TurboID-based EGFR-FabID was attempted, we could not purify TurboID-fused cells from the culture media of Expi293F cells. As shown in Fig. 1, AirID-based AGIA-FabID was purified and had high proximity biotinylation activity. Taken together, these results suggest that AirID is suitable for fabricating Fab-fusion molecules.

We performed flow cytometry to determine whether EGFR-FabID recognises EGFR on the cell surface and biotinylates proteins on the plasma membrane. Expi293F cells stably expressing EGFR were biotinylated with EGFR-FabID for 2 h, stained with STA-Alexa488, and its fluorescence was measured using FACS (Supplementary Fig. 2a). The results showed an increase in the population of cells biotinylated on the cell surface by treatment with EGFR-FabID and biotin. This suggested that EGFR-FabID biotinylated the cell surface after binding to EGFR. An MTS assay using A431 cells was also performed to verify the cytotoxicity of EGFR-FabID (Supplementary Fig. 2b). The results of the MTS assay confirmed that addition of EGFR-FabID or AGIA-FabID did not affect cell survival.

Next, to confirm whether biotinylation by EGFR-FabID occurs in the extracellular region, biotinylated peptides were purified by using Tamavidin 2-REV and analysed by LC-MS/MS according to previous reports[38,39]. MS analysis indicated that EGFR-FabID biotinylated five and seven lysine residues in the extracellular and intracellular regions of EGFR, respectively (Fig. 2f). Furthermore, since the anti-EGFR antibody recognises an epitope around Ser380 in the extracellular region[34] and the 3D structure of the extracellular region of EGFR has been reported (PDB code: 1IVO [https://doi.org/10.2210/pdb1ivo/pdb])[36], we estimated the distance between the binding site of EGFR-FabID and biotinylation sites. These five biotinylation sites in the extracellular region were localised within 5 nm of the epitope (Fig. 2g). These interval values seem reasonable because the biotinylation range of the BioID enzyme is presumably ~10 nm[13], strongly suggesting that EGFR-FabID binds to the epitope and subsequently biotinylates it. Taken together, these results indicated that EGFR-FabID functions as a specific proximity biotinylation probe in the extracellular region.

**Proximity exPPI analysis of EGFR by EGFR-FabID on suspension culture Expi293F cells overexpressing EGFR transiently**
As shown in Fig. 2, EGFR-FabID catalysed extracellular biotinylation of EGFR by incubation with serum-free medium suspension of Expi293F cells transiently overexpressing EGFR. EGFR reportedly interacts with other membrane proteins[32,33]. LC-MS/MS analysis to determine the biotinylation of other proteins revealed that EGFR-FabID biotinylated 639 peptides in transiently EGFR-overexpressing Expi293F cells

(Fig. 3a). Further, when AGIA-FabID was used as a negative control, 189 peptides were significantly biotinylated by EGFR-FabID ($P < 0.05$ and EGFR-FabID/AGIA-FabID Ratios >1 in Supplementary Data 1). Since our detection method using Tamavidin 2-REV could identify biotinylation sites on proteins[38,39], we analysed whether the biotinylation sites were localised in the extracellular regions using TMHMM (https://services.healthtech.dtu.dk/service.php?/TMHMM-2.0/)[40] and Uniprot (https://uniprot.org) databases. Statistically predominant biotinylation sites of 64 peptides were found in the extracellular region (red dots in Fig. 3b), and the remaining 125 peptides were localised in intracellular or unknown cellular regions (black dots, Supplementary Data 1).

Further analysis focused on the 22 proteins that were biotinylated in their extracellular region because EGFR-FabID recognised the extracellular epitope. Based on the IntAct database, which includes EGFR-interacting proteins (https://ebi.ac.uk/intact/)[41], 12 proteins were known to interact with EGFR (see black characters in Fig. 3c). Twelve out of 22 proteins (55%) showed high follow-up capability with recorded evidence, suggesting that the remaining 10 proteins were also unknown interacting or proximal proteins (see blue characters in Fig. 3c). To compare the proximal exPPIs with the existing PPI database, these interactions were analysed using the Drug Target Excavator (DTX) (https://harrier.nagahama-i-bio.ac.jp/dtx/). This database depicts human protein/metabolite interaction pathways between a given pair of proteins. For DTX analysis, a direct protein-protein interaction pathway is presented as one edge. If a protein-protein interaction pathway had two edges, it indicated that the protein interaction was mediated by another protein. Proximal exPPIs detected by EGFR-FabID were found within three edges (see the DTX edge in Fig. 3c). In addition, pathway analysis of the 22 proteins showed individual interactions with EGFR rather than multiple interactions, forming a large network (Fig. 3d). Taken together, these results indicate that EGFR-FabID provides a proximity interactome of extracellular regions between EGFR and other membrane proteins in transiently EGFR-overexpressing suspended cells.

**Biotinylation analysis of endogenous EGFR by EGFR-FabID in human A431 cells**
Next, we attempted to determine whether EGFR-FabID works on endogenous EGFR located on the surface of adherent cells. To select an adherent cell type, the protein level of endogenous EGFR and the reactivity of EGFR-FabID were investigated using immunoblotting and STA-PDA. The anti-EGFR (clone EMab-134) reacted with the EGFR protein in eight cell lines (Supplementary Fig. 2c). The expression level of EGFR protein was the highest in human squamous carcinoma epithelial cells (A431), whereas it was not detected in HEK293T and Expi293F cells. Comparison of EGFR expression indicated that EGFR protein expression in A431 cells was higher than that in stably EGFR-overexpressing Expi293F cells and similar to that in transiently EGFR-overexpressing Expi293F. To validate the performance of EGFR-FabID, we compared five cell lines with different EGFR expression levels: A431 (high), Expi293F-EGFR stable (middle), HeLaS3 (low), NCI-H1975 (low), and NCI-H226 (low) cells. Immunoblotting showed that EGFR biotinylation by EGFR-FabID was found in all lysates (red arrowheads in Fig. 4a); this was confirmed using STA-PDA. In addition, immunostaining indicated that EGFR-FabID

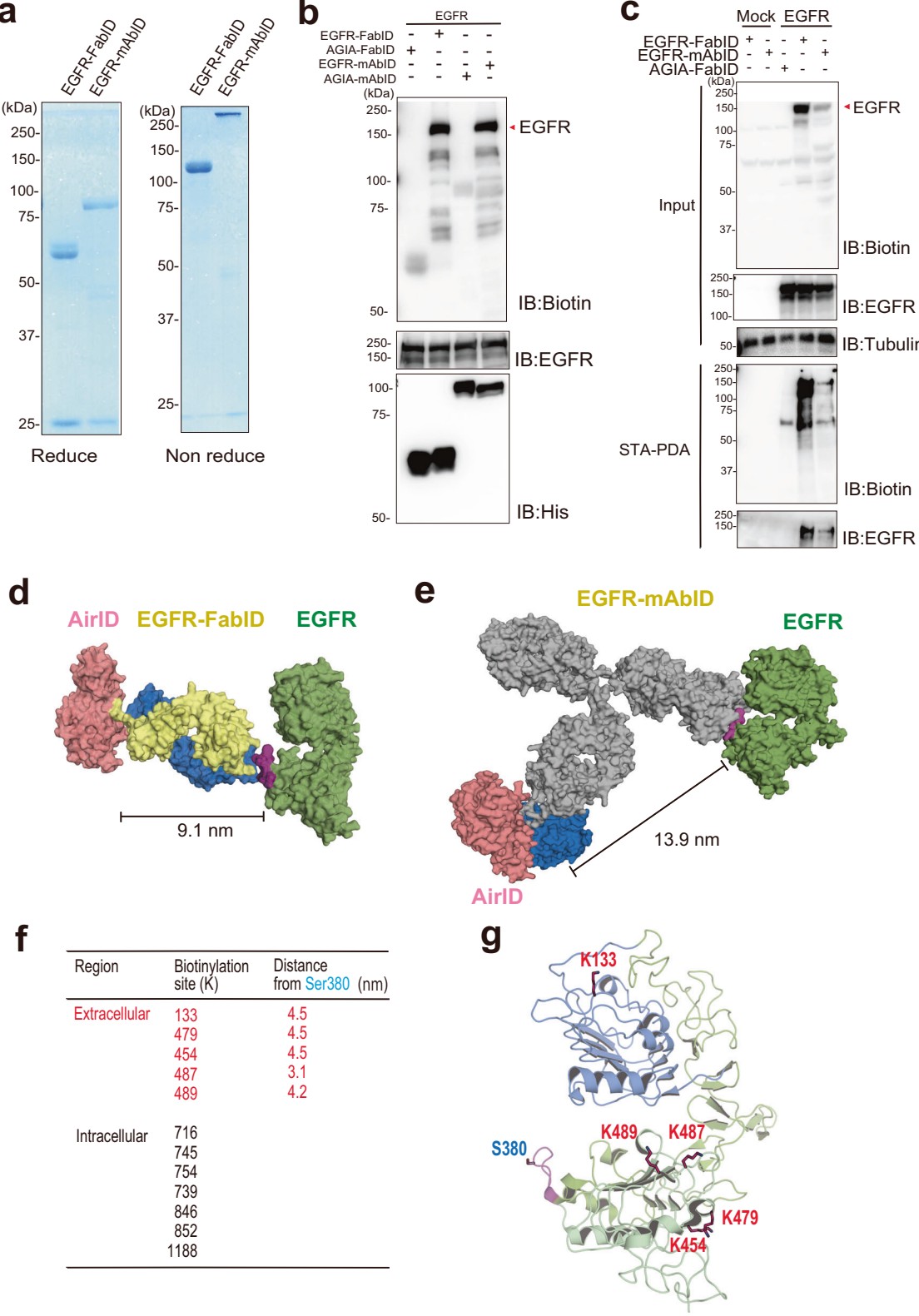

**f**

| Region | Biotinylation site (K) | Distance from Ser380 (nm) |
|---|---|---|
| Extracellular | 133 | 4.5 |
| | 479 | 4.5 |
| | 454 | 4.5 |
| | 487 | 3.1 |
| | 489 | 4.2 |
| Intracellular | 716 | |
| | 745 | |
| | 754 | |
| | 739 | |
| | 846 | |
| | 852 | |
| | 1188 | |

biotinylated EGFR on the extracellular region of the cell membrane in A431 cells (Fig. 4b), whereas AGIA-FabID did not. These results suggest that EGFR-FabID biotinylates endogenous EGFR in cells at low and high expression levels. We next performed experiments using the well-known endocytosis inhibitor Dynasore to check that EGFR-FabID works under conditions in which endocytosis does not occur. After treatment of non-small-cell lung cancer-derived NCI-H226 cells with Dynasore, the cells were biotinylated with EGFR-FabID (Supplementary Fig. 2d). The immunoblotting of cell lysates showed that EGFR was biotinylated even in the endocytosis-inhibited cells and the band intensity was similar between treatments with and without Dynasore. This indicates that endocytosis is not essential for biotinylation of EGFR-FabID and the majority of biotinylated EGFR is present at the cell surface.

**Fig. 2 | Biotinylation analysis of EGFR by EGFR- FabID on the cell membrane.**
**a** SDS-PAGE of EGFR-FabID and EGFR-mAbID synthesised by the Expi293F expression system, purified by Ni Sepharose in reducing and non-reducing states, and stained with CBB. **b** Biotinylation of EGFR using EGFR-FabID and EGFR-mAbID in vitro. EGFR was synthesised using a wheat cell-free protein synthesis system-based Disulfide Bond PLUS Expression Kit. **c** Immunoblot analysis of EGFR biotinylated by EGFR-FabID and EGFR-mAbID on the plasma membrane using EGFR-overexpressing Expi293F cells. **a**–**c** The experiment was independently repeated twice, with similar results. **d** EGFR-FabID modelling diagram calculated from structural information and AlphaFold. Green represents EGFR, and yellow and blue represent anti-EGFR Fab. Pink represents AirID. **e** EGFR-mAbID modelling diagram calculated from structural information and AlphaFold. Green represents EGFR; grey, and blue represent anti-EGFR mAbs. Pink represents AirID. **f** Biotinylation site of EGFR detected by LC-MS/MS; the intermolecular distance between S380 of EMab-134 epitope and biotinylated lysine residue was measured by PyMOL based on structural information (PDB 1IVO [https://doi.org/10.2210/pdb1igt/pdb]) and (**g**), biotinylation sites of extracellular domain EGFR. Source data are provided as a Source Data file.

Many researchers have used A431 cells for EGFR analysis because the EGFR gene has been cloned from this cell line[42]. In addition, EGFR-FabID clearly biotinylated the endogenous EGFR protein in A431 cells (Fig. 4b). Based on these results, further analysis of exPPI interactions using EGFR-FabID on adherent cells was focused on A431 cells.

Previous structural analysis data by Ogiso et al. suggested that site 1 in domain 1 and sites 2 and 3 in domain 3 were important for EGF recognition by EGFR[36]. EGFR recognition site 2 in EGFR-FabID was located between amino acids 377–386 in domain 3. We examined the effect of FabID fusion on EGF/EGFR binding because the EGFR-FabID recognition site was located at site 2. We examined whether the addition of EGFR-FabID affected the EGFR ligand response using gefitinib. In A431 cells, after EGF ligand treatment with or without gefitinib, EGFR-FabID was supplied to the cells. Using the phospho-EGFR specific antibody, immunoblotting indicated that the kinase activity of EGFR dramatically increased by the treatment of EGF ligand in the presence of EGFR-FabID (Fig. 4c) and was completely inhibited by gefitinib treatment. Notably, the large β-sheets at sites 1 and 3 of EGFR were important for the interaction between EGFR and the EGFR ligand TGF-α, which had a structure similar to that of EGF[37]. Binding of EGFR-FabID to site 2 did not decrease EGFR phosphorylation, suggesting that the EGF-EGFR complex was stable during the binding of EGFR sites 1 and 3 to EGF. These reports would support the understanding of the results that EGFR-FabID did not affect EGFR activity after treatment with the EGF ligand.

Considering these responses of EGFR in A431 cells, biotinylation of EGFR by EGFR-FabID was attempted under three conditions, with or without EGF ligand and with gefitinib treatment in the presence of EGF. Biotinylation of EGFR was analysed using LC-MS/MS, as shown in Fig. 2. Interestingly, three biotinylated peptides on K454, K479, and K487 or K489 were found under all conditions, and the K489-biotinylated peptide increased on treatment with EGF ligand (red dots in Fig. 4d). Two biotinylated peptides on K479 and K487 or 489 increased on treating EGFR with gefitinib and EGF ligand (Fig. 4e), indicating that biotinylation sites changed in the presence or absence of EGF ligand or the presence of a kinase inhibitor. These results suggest that K489 or K479 biotinylation could be a marker of EGF binding or gefitinib treatment of EGFR using EGFR-FabID, respectively.

As shown in Fig. 2, biotinylation sites of EGFR from transiently EGFR-overexpressing Expi293F cells were analysed and compared with biotinylation sites on endogenous EGFR in A431 cells. In the extracellular region, almost the same pattern of biotinylation sites was observed (Supplementary Fig. 3a). Interestingly, the intracellular region of endogenous EGFR in A431 cells was not biotinylated by EGFR-FabID. However, in transiently EGFR-overexpressing Expi293F cells, the intracellular region was biotinylated at seven sites (black dots in Fig. 4c and Supplementary Fig. 3b), suggesting that transiently overexpressed EGFR proteins contain a mixture of forms that differ in conformation from endogenous EGFR proteins on the plasma membrane. We performed MS analysis of Expi293F stably expressing EGFR cells to determine whether transient overexpression of EGFR causes biotinylation of the intracellular domain of EGFR by EGFR-FabID (Supplementary Data 2). We found no biotinylation of the intracellular domain of EGFR in stably expressing Expi293F cells. All biotinylated

lysine residues were in the extracellular domain of EGFR (Supplementary Fig. 3a). This result suggested that the intracellular domain of EGFR was biotinylated due to its transient overexpression. Taken together, these results indicate that EGFR-FabID profiling of biotinylation sites on the extracellular region could help understand the various conformations of EGFR on the cell membrane.

## Total proximity exPPI analysis using EGFR-FabID among three cell lines

The biotinylation sites of the EGFR protein were detectable in adherent A431 cells (Fig. 4). Therefore, to evaluate the potential versatility of EGFR-FabID in adherent cells and the commonality of proximity proteins of EGFR, the human lung squamous cell carcinoma cell line NCI-H226 was also used for EGFR exPPI analysis as a new adherent cancer cell line. Biotinylated proteins from A431 and NCI-H226 cells, with or without EGF ligand and with gefitinib treatment in the presence of EGF, were analysed (Fig. 5a, Supplementary Data 3, 4). LC-MS/MS analysis revealed 481 and 849 biotinylated peptides from A431 and NCI-H226 cells, respectively, under the three treatment conditions. The biotinylation site in the intracellular region of EGFR was also not found in NCI-H226 cell lines expressing endogenous EGFR (Supplementary Fig. 3c), similar to A431 cells. The location of biotinylation sites was identified using TMHMM (http://www.cbs.dtu.dk/services/TMHMM/) and Uniprot (https://uniprot.org) databases; 216 and 347 biotinylated peptides from A431 and NCI-H226 cells, respectively, were annotated as peptides on the cell surface (red characters in Fig. 5a and red dots in Fig. 6a). The remaining peptides were localised in intracellular or unknown cellular regions (Supplementary Data 3, Supplementary Data 4). Finally, of these extracellular biotinylated peptides, 44 and 66 proteins, including EGFR, were found to be membrane proteins showing proximal exPPI with EGFR in A431 and NCI-H226 cells, respectively (Supplementary Fig. 4a). The DTX analysis revealed 15 and 16 proteins from A431 and NCI-H226 cells, respectively, interacting with EGFR. The remaining 29 and 50 proteins from the A431 and NCI-H226 cells, respectively, were unknown proximal exPPI candidates (denoted by blue numbers and protein names in Fig. 5a and Supplementary Fig. 4a).

Commonality analysis was performed based on the biotinylated protein data from the three cell lines (EGFR-overexpressing Expi293F, A431, and NCI-H226). Except for EGFR, five proteins: EEF1A1 (elongation factor 1-alpha 1), ENO1 (alpha-enolase), MICA (MHC class I polypeptide-related sequence A), INSR (insulin receptor), and CD44 (cell differentiation 44), were the most common proximal exPPI proteins, of which EEF1A1[33] and ENO1[43] were known EGFR interactors (see black characters in Fig. 5b), and MICA, INSR, and CD44 were unknown ones (see blue characters). Surprisingly, the translation elongation factor EEF1A1 was biotinylated by EGFR-FabID. However, the non-cytotoxicity of EGFR-FabID was confirmed by MTS (Supplementary Fig. 2b). This suggested that the biotinylated EEF1A1 detected was not that of lysed cells. Although EEF1A1 was a translation elongation factor, it has also been found on the plasma membrane[44]. Then we examined the co-localisation of EGFR with EEF1A1 that was biotinylated in all of the three cell lines (Supplementary Fig. 10b). Immunostaining without permeabilization confirmed the co-localisation of EEF1A1 and EGFR on

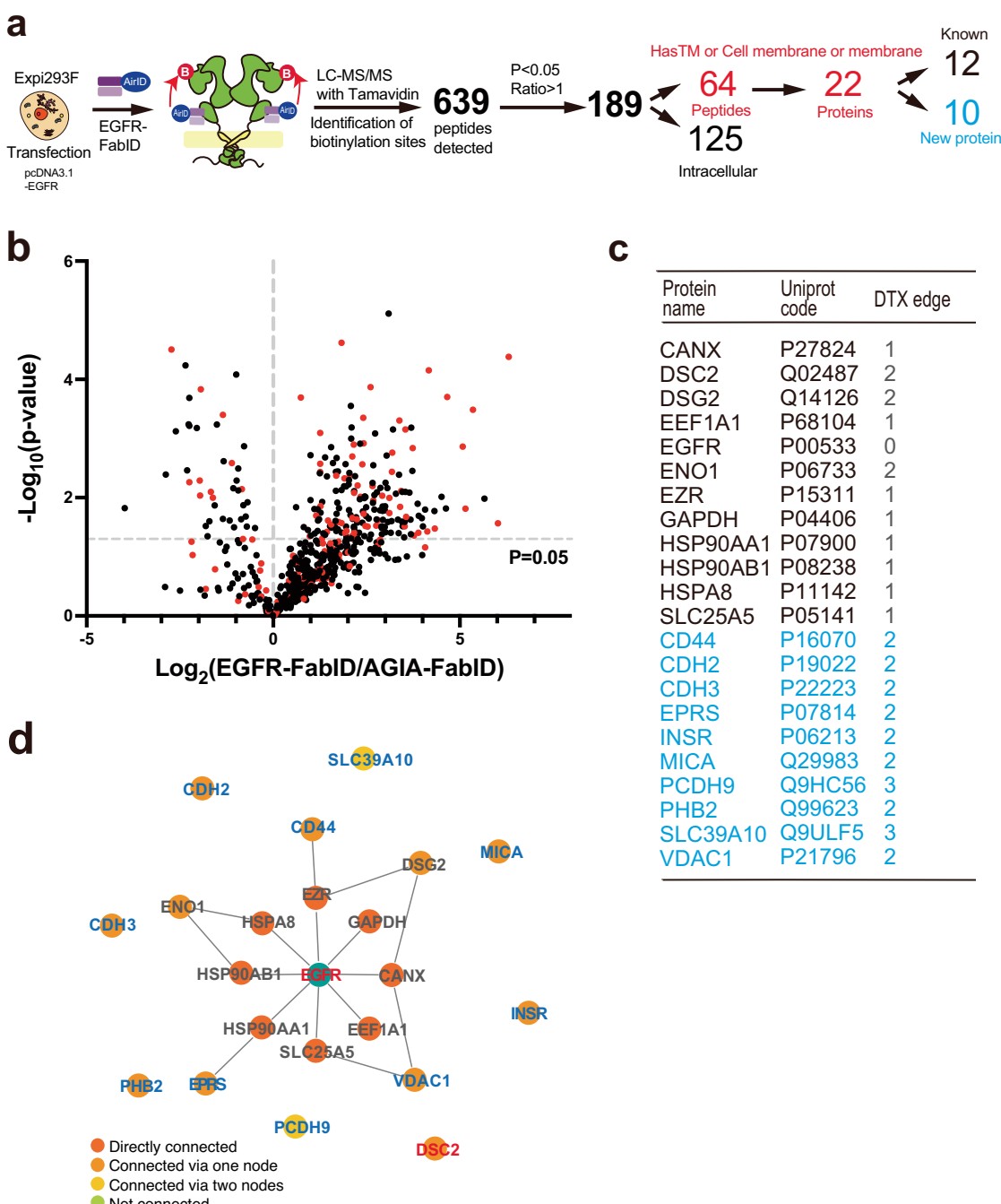

**Fig. 3 | Proximity extracellular interactome of over-expressing EGFR by EGFR-FabID on cells. a** Workflow of PPI analysis targeting EGFR overexpression. LC-MS/MS identified 640 biotinylated peptides, of which 189 were selected as those with AGIA-FabID ratios >1 and *P* < 0.05. The number of peptides was then narrowed to 64 peptides that had transmembrane (TM) regions or contained "membranes" in the gene ontology (GO) term. 64 peptides were matched into 22 proteins. Twelve of the 22 proteins were known to interact with EGFR, and 10 were novel EGFR-interacting proteins. **b** Volcano blot (3 biological replicates) of peptides detected as biotinylated peptides by LC-MS/MS in (**a**). Peptides derived from cell membrane proteins are indicated by red dots. **c** Table of extracellular proteins identified by mass spectrometry (three biological replicates, EGFR-FabID/AGIA-FabID ratios >1 and

the cell membrane. *P* < 0.05). The number of DTX edges for each protein and EGFR are shown in the table. Proteins in black font are those already known to interact with EGFR, whereas those in blue represent unknown EGFR-interacting proteins. **d** Pathway analysis of extracellular proteins detected using mass spectrometry. The Gene Ontology software Drug Target Excavator (DTX) (https://harrier.nagahama-i-bio.ac.jp/dtx/) and IntAct database, including EGFR-interacting proteins (https:///ebi.ac.uk/intact/), were used to analyse protein interactions. **b**, **c** Significant changes in the volcano plots and heat map were calculated by Student's two-sided *t* test and the false discovery rate (FDR)-adjusted *P*-values calculated using Benjamini–Hochberg method are shown in the Supplementary Data 1.

the cell membrane. These results suggested that EGFR-FabID biotinylates EEF1A1 on the plasma membrane. In addition, 15 common proteins were found between the A431 and NCI-H226 cell lines and consisted of 6 known and 9 unknown proteins. Although Expi293F cells overexpressing EGFR are floating cells and the remaining two cell lines, A431 and NCI-H226, are adherent cells, common proteins, including known interactors, were found between each cell, indicating that EGFR-FabID has similar functions in both adherent and non-adherent cells. The remaining 8, 19, and 41 proteins were independently observed in each cell line, suggesting that exPPIs with EGFR may

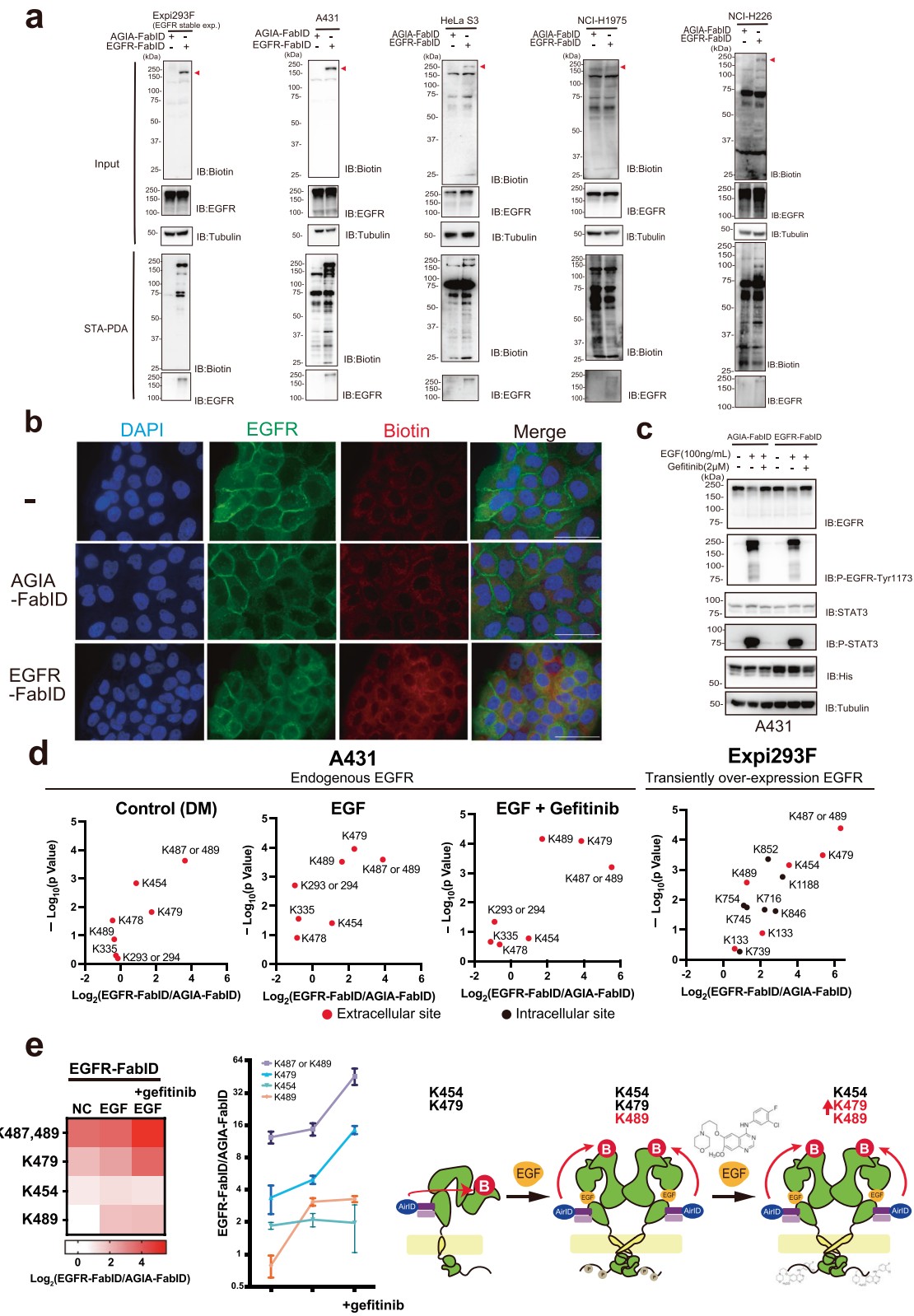

differ by cell type. The proximity exPPI results of the three cell lines with EGFR-FabID revealed many known interacting proteins, suggesting that EGFR exPPI may occur in complex with known interacting proteins.

To further compare the proximity exPPIs with the existing PPI database, these interactions were analysed using DTX [https://harrier.nagahama-i-bio.ac.jp/dtx/)]. Interaction network diagrams of

biotinylated proteins in A431 and NCI-H226 cells showed that many proteins interacted with EGFR within three nodes (Fig. 5c). In A431 cells, all but DSG3 interacted with EGFR within two nodes. Some biotinylated proteins also interacted with each other. In NCI-H226 cells, all proteins, except L1CAM, interacted with EGFR within two nodes. The interaction network diagram, including proteins mediating interactions with EGFR, also confirmed that some biotinylated proteins

**Fig. 4 | Biotinylation analysis of endogenous EGFR by EGFR-FabID in human A431 cells. a** Streptavidin pull-down assay of endogenous EGFR on the plasma membrane biotinylated by EGFR-FabID in various cells. Red arrowheads indicate the molecular weight of EGFR. **b** Immunostaining with EGFR-FabID in A431 cells. Control-treated areas included those without FabID treatment and those treated with AGIA-FabID. Scale bar represents 50 μm. **c** Kinase assay using A431 cells. Phosphorylation of EGFR and STAT3 activated by EGF stimulation was then confirmed by immunoblotting. **a–c** The experiment was independently repeated twice, with similar results. **d** Volcano plot of biotinylated EGFR peptides identified by mass spectrometry (three biological replicates). Red dots represent peptides derived from the extracellular domain of EGFR, and black dots represent peptides derived from the intracellular domain. Significant changes in the volcano plots and heat map were calculated by Student's two-sided *t* test and the false discovery rate (FDR)-adjusted *P*-values calculated using Benjamini–Hochberg method are shown in the Supplementary Data 1, 3. **e** Heat map showing changes in biotinylated peptide sites upon EGF and gefitinib treatment of A431 cells (three biological replicates). Schematic of the changes in the EGFR biotinylated site. Error bars represent standard deviations. Source data are provided as a Source Data file.

mediate common molecules in the three cell lines (Supplementary Fig. 4b). Taken together, these results indicate that this approach using EGFR-FabID could be used to analyse the proximal extracellular interacting proteins of EGFR on the cell membranes of both adherent and floating cells.

The EGF ligand was not detected by mass spectrometry as a biotinylated peptide in A431 and NCI-H226 cells. To understand why EGF was not biotinylated by EGFR-FabID, a model illustration of EGFR-FabID bound to the EGFR-EGF complex was generated using its crystal structure and AlphaFold (Supplementary Fig. 5a). The proximity biotinylation enzyme biotinylated accessible lysines that were exposed on the protein surface by releasing short-lived, hydrophilic biotinoyl-5′-AMP. The amino acid sequence of the EGF ligand contained lysine residues at positions 28 and 48. The model figure suggested that K48, which was located inside EGF, was not easily accessed by the proximity biotinylation enzyme. Furthermore, K28 was located at the boundary between the light chains of EGFR-FabID and EGFR, which was also potentially difficult for hydrophilic biotinoyl-5′-AMP to access. We then examined whether EGFR-FabID could biotinylate the ligand if lysine residues are present on the surface of the ligand. To ensure lysine sites for biotinylation, we used the FLAG-GST (glutathione *S*-transferase) tag was used because it has totally 23 lysine residues. We then constructed a vector of FLAG-GST-EGF where a TEV-His-AGIA-GS linker inserted between FLAG-GST and EGF. The FLAG-GST-EGF ligands induced the phosphorylation of EGFR (IB: P-EGFR1173 in Supplementary Fig. 5b), indicating the activity as a ligand. The cells were cultured for two days after transfection with this vector and biotinylated by EGFR-FabID. The culture medium was analyzed by the immunoblotting after GST pull-down by glutathione-sepharose beads, and biotinylation of FLAG-GST-EGF ligand was detected (Supplementary Fig. 5c). This result showed that EGFR-FabID can biotinylate the ligand if there are sufficient lysine residues on the ligand surface and suggested that no biotinylation of EGF by EGFR-FabID was due to the lack of exposed lysine residues on the ligand surface.

## Dynamic responses of proximity exPPIs on treatment with EGF ligand with or without gefitinib

In the EGF signalling response, dimeric EGFR phosphorylate each other at tyrosine residues in the intracellular region to form a signal complex. Gefitinib is a highly specific tyrosine kinase inhibitor of EGFR[27]. To understand the effect of gefitinib on exPPI, we compared the proximal exPPIs in A431 and NCI-H226 cells in the presence of EGF ligand (EGF) and EGF plus gefitinib (EGF+gefitinib) with a control without ligand (DMSO). From each treatment of the two cell lines, proximity exPPI analysis of EGFR yielded approximately 30 and 35–50 proteins, respectively (Fig. 6a, Supplementary Fig. 6). These proximity exPPI proteins were analysed using a heatmap and scatter plot based on protein data from LC-MS/MS (Supplementary Data 5 and 6). In both A431 and NCI-H226 cells, 20 proteins were common proximity exPPIs (Fig. 6b, Supplementary Fig. 7) (HLA-C was removed from the common proteins because no values were detected in the protein MS data for A431 cells.), and the remaining 23 and 46 proteins were specific to A431 and NCI-H226 cells, respectively (Supplementary Fig. 8a, b). Interestingly, EGFR-FabID-driven biotinylation was dramatically increased or

decreased by EGF treatment compared with that by DMSO treatment. Furthermore, gefitinib treatment in the presence of EGF (EGF+gefitinib) also affected biotinylation of the proximal exPPIs with EGFR, suggesting that treatment with EGF ligands and tyrosine kinase inhibitors changes the proximity exPPIs of EGFR. Additionally, different cells exhibited different patterns of biotinylation induced by EGFR-FabID in the presence of EGF and/or gefitinib.

EGFR-FabID detected exPPI changes of EGFR after EGF stimulation in the presence and absence of Gefitinib treatment (Fig. 6a, b). Gefitinib binds to the ATP binding site of EGFR and suppresses its phosphorylations in the presence of EGF[45,46]. Testing of gefitinib binding to EGFR in the absence of EGF could lead to a better understanding of the effects of Gefitinib. Therefore, to confirm the biotinylation of EGFR exPPI proteins with gefitinib treatment alone, we attempted MS analysis and identified proximity exPPI proteins using EGFR-FabID in gefitinib treatment (Supplementary Fig. 9a, b, Supplementary Data 7 and 8). Interestingly proteins other than those biotinylated during EGF-stimulation or EGF-stimulated gefitinib treatment were also biotinylated to EGFR-FabID in the gefitinib treatment alone (Supplementary Figs. 6a, 9b). These results suggest that even in the absence of ligand, drugs that bind to the intracellular region of RTKs have a potential to modify the interaction of the extracellular region.

To confirm the presence of exPPIs with EGFR, we attempted to biochemically detect the interaction between the extracellular region of EGFR using the AlphaScreen method. PTK7 (tyrosine receptor kinase 7), INSR, and ADAM17 (disintegrin and metalloproteinase domain-containing protein 17), commonly found in proximity exPPIs with EGFR-FabID in both A431 and NCI-H226 cells, were selected. PTK7 and INSR share downstream signalling with EGFR[47–49], although they are not known as EGFR interactors. ADAM17 is also a major protease of the EGF family[50]. For the biochemical interaction of these proteins with the extracellular domain of EGFR, the extracellular region of EGFR was synthesised by a wheat cell-free system to form S-S complexes, and the extracellular interactions were analysed using the AlphaScreen method. Since the alpha subunit of INSR is known as the extracellular insulin-binding site[51], it was used in the assay. The AlphaScreen method showed that PTK7, INSR alpha, and ADAM17 significantly bound to the extracellular domain of EGFR (Fig. 6c). Furthermore, the interaction of PTK7 and INSR with EGFR was also confirmed using the NanoBiT method, which allows the measurement of interacting proteins in cultured cells (Fig. 6d). However, ADAM17 did not work on the Nano-BiT method. We also performed immunostaining for INSR, ADAM17, and PTK7 to examine their co-localisation with endogenously expressed EGFR. ADAM17 and PTK7 co-localised with EGFR in A431 cells (Supplementary Fig. 10a). The results showed that endogenous EGFR co-localised with endogenous ADAM17 and PTK7 on the plasma membrane of A431 cells. INSR was also confirmed to colocalize with EGFR on the plasma membrane of NCI-H226 cells (Supplementary Fig. 10b). These biochemical and cellular interaction analyses suggested that INSR, ADAM17, and PTK7 interacted with the extracellular region of EGFR.

Next, the change in proximity exPPIs in EGF, with or without gefitinib treatment, was validated by proximity ligation assay (PLA) method in A431 cells. As shown in Fig. 6b (A431 cell panel), proximity

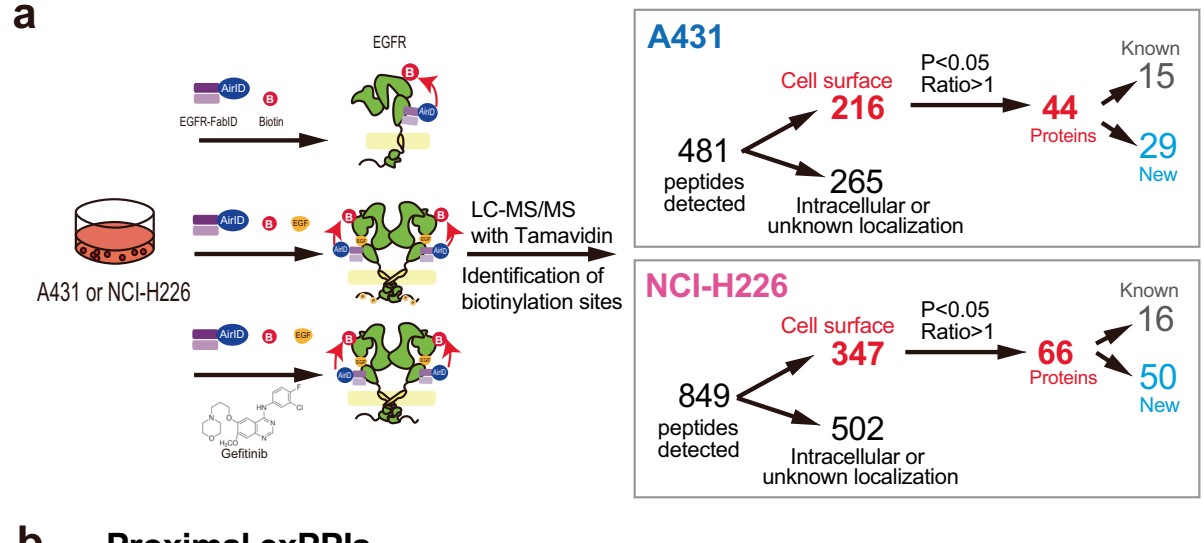

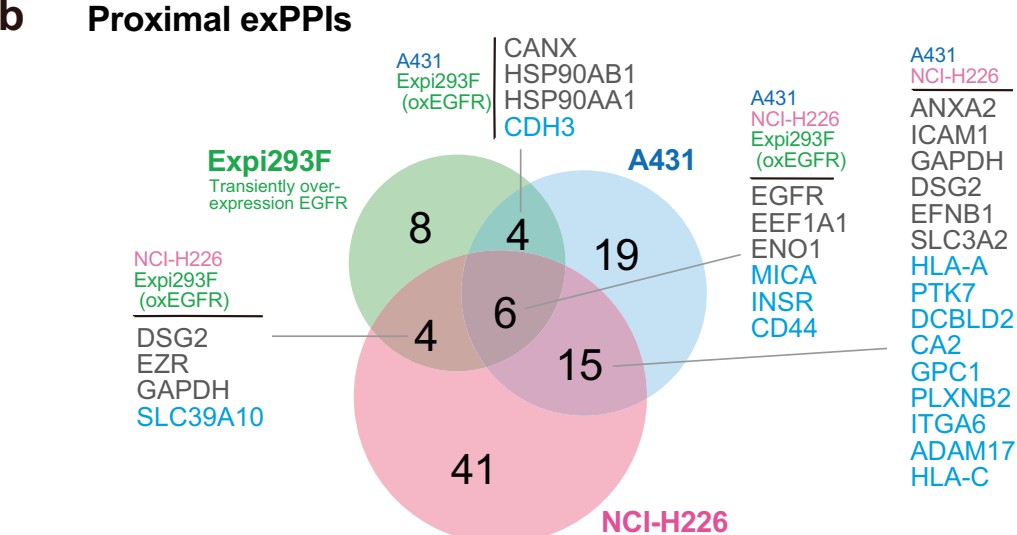

## b  Proximal exPPIs

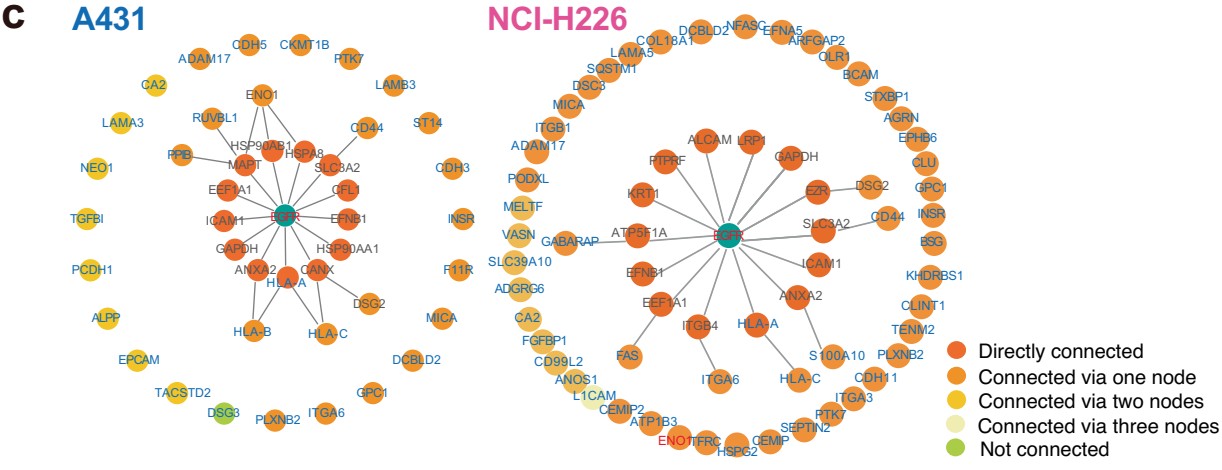

exPPI of PTK7 with EGFR was increased by gefitinib treatment. A large number of PLA signals were detected in PTK7, and the PLA signal was also reduced by EGF+gefitinib treatment compared with that by EGF treatment. (Fig. 6e). The PLA signal values and biotinylation of PTK7 were mismatch. This is probably because the PLA signal of PTK7 was so high even before treatment with ligand or drug that it exceeded the threshold for the PLA method to detect interaction changes upon addition of drug or ligand. In addition, not all interaction detections coincided because EGFR-FabID and PLA have different distances of detectable interaction. In contrast, the proximity exPPI of INSR was increased by EGF and EGF+gefitinib (Fig. 6b, A431 cell panel), and the PLA signal showed a significant increase (INSR in Fig. 6e). Furthermore, the proximity of exPPI of ADAM17 with EGFR also increased by EGF +gefitinib treatment (Fig. 6b, A431 cell panel), and the PLA signal also

**Fig. 5 | Total proximity extracellular biotinylation analysis of EGFR using EGFR-FabID in A431 and NCI-H226 cells with or without EGF ligand and/or gefitinib. a** Workflow of PPI analysis targeting A431 cells. The same procedure as that in Fig. 3 was used to narrow down the peptides to extracellular peptides. As a result, 216 peptides in A431 cells and 347 peptides in NCI-H226 cells were derived from cell surface peptides. In A431 cells, 15 molecules were known to interact with EGFR, and 29 molecules were unknown interacting proteins. In NCI-H226 cells, 16 molecules were known to interact with EGFR, and 50 molecules were unknown interacting proteins. **b** Venn diagram of proteins predominantly biotinylated in EGFR-overexpressing Expi293F, A431, and NCI-H226 cells. The numbers represent the number of proteins in the region (EGFR-FabID/AGIA-FabID ratios >1 and $P < 0.05$). Significant changes in the volcano plots and heat map were calculated by Student's two-sided $t$ test and the false discovery rate (FDR)-adjusted $P$-values calculated using Benjamini–Hochberg method are shown in the Supplementary Data 3, 4. **c** Pathway analysis of 44 proteins in A431 cells and 66 proteins in NCI-H226 cells, obtained from workflow shown in Fig. 5a.

increased in the EGF+gefitinib treatment (ADAM17 in Fig. 6e). These results indicated that PLA signals were similar to biotinylation signals from EGFR-FabID proximity exPPIs with EGF. Taken together, these results strongly suggest that EGF ligand and a kinase inhibitor provide dynamic exPPIs of EGFR on the cell membrane, and the FabID system can detect them via biotinylation.

## Discussion

In this study, we developed the EGFR-FabID and EGFR-mAbID probes, and then EGFR-FabID was used to study the exPPI activity of EGFR. The structural model suggested that the distance from FabID to the antigen was ~10 nm (Fig. 2d) and that from mAbID to the antigen would be ~14 nm (Fig. 2e), resulting in a higher biotinylation activity of EGFR-FabID than that of mAbID (Fig. 2c), although the in vitro biotinylation activity was almost the same (Fig. 2b). These results suggest that the distance between the probe and target protein is a key factor in proximity biotinylation of exPPI partners. Recently, EMARS[52,53] and μMap[10] were reported to detect proximal interactions with target membrane proteins. These methods used a full-size IgG antibody and not the Fab form. Furthermore, horseradish peroxidase (HRP) and a chemical molecule, ({Ir[dF(CF$_3$)ppy]$_2$(dtbbpy)}PF$_6$), are used as biotinylation catalysts in EMARS and μMap, respectively. Biotinylation via HRP with biotin phenol occurs in the range of 200–300 nm, while μMap has a biotinylation range of 50–100 nm[11]. In addition, the chemocatalytic molecule used in μMap is conjugated on a secondary antibody but not the primary antibody[10], suggesting that the distance between the biotinylation catalyst and target membrane protein is even further. FabID conjugates the AirID enzyme directly to the C-terminus of Fab. Taken together, the FabID system can efficiently biotinylate interacting proteins in the proximity of membrane proteins, thus making it the ideal method for the analysis of exPPIs on the cell membrane.

BioID-based exPPI analysis has been performed by expressing proteins with BioID genetic insertion of BioID enzyme between the domains of the extracellular regions of membrane proteins[26]. Although this approach has revealed important exPPI results, it has several drawbacks. Many membrane proteins have a signal peptide at their N-terminus that is required for membrane-spanning and prevents insertion of the BioID enzyme to the N-terminus. This necessitates that membrane proteins with an N-terminus exposed to the extracellular space integrate BioID between domains in the extracellular region. However, the integration of BioID enzymes between the domains of membrane proteins is likely to affect the exPPI of these proteins due to structural effects. In addition, BioID-fused proteins do not allow for exPPI analysis of endogenously expressed membrane proteins in their native form. In this regard, FabID could be used to analyze the endogenously expressed membrane protein exPPI, although it requires well-characterised antibodies against the target protein. EGFR-FabID was used to analyze the exPPIs of endogenously expressed EGFR (Fig. 5). The advantage of the FabID method is that it can analyse the proximity exPPI of endogenous membrane proteins, which are difficult to analyse using conventional BioID enzyme-based methods.

Using an EGFR-FabID system based on AirID to analyse exPPIs of the EGFR protein in living cell lines (both adherent and nonadherent cells), we revealed biotinylation sites by LC-MS/MS using Tamavidin2-

REV[38,39] and indicated that proximity biotinylation occurred in the extracellular region. The analysis, therefore, provided proximity exPPIs with EGFR and found many well-known EGFR interactors, as well as unknown proximity exPPI candidate proteins (Fig. 5). Individual analyses also indicated direct exPPIs with the extracellular region of EGFR (Fig. 6). Because other conventional methods have not identified biotinylation sites, there is no information on the proximal region of the identified proteins. These results indicate that biotinylation sites provide important information for interactome analysis of membrane proteins.

The FabID system showed that exPPI with EGFR changes dynamically with the addition of EGF ligand and gefitinib (Fig. 6 and Supplementary Fig. 8). EGF ligands induce dimerisation of EGFR along with other conformational changes[36]. The EGFR-FabID method detected a change in biotinylation sites on EGFR with or without EGF (Fig. 4d, e, and Supplementary Fig. 3), contributing to the dynamic influence of proximity exPPIs with EGFR on the cell membrane. As EGF induces conformational changes in the extracellular region of EGFR, it is consistent that EGF affects the interaction of exPPIs with EGFR. In addition, gefitinib inhibits tyrosine kinase activity, resulting in complete loss of phospho-tyrosine sites[45]. Since phospho-tyrosine is used as a scaffold site for the formation of the EGF signal complex[54], gefitinib may also inhibit its complex formation. The results of EGF+gefitinib treatment in EGFR-FabID, therefore, suggest that the inhibition of EGFR complex formation in the intracellular region affects the interaction with extracellular regions of EGFR. Taken together, FabID provides a tool for extracellular interaction analysis, as it can capture not only the interaction analysis of extracellular regions but also interacting proteins altered by ligands and drugs.

The EGFR-FabID method revealed direct interactions in the extracellular domain of INSR and EGFR, which interact during intracellular signalling[48,49]. This finding may indicate that extracellular signalling interactions occur between heterogeneous receptors on the membrane. Further, the exPPI analysis of various RTKs using FabID on the cell membrane may provide new insights into signalling cooperation in the extracellular region. Furthermore, since they can change with drug treatment, FabID-based exPPIs may provide a new perspective on drug development for RTKs.

In conclusion, the FabID system for the analysis of extracellular interactions can capture not only the interaction analysis of extracellular regions but also interacting proteins altered by ligands and drugs, thus delivering a powerful tool for the study of membrane proteins.

## Methods

### Reagents
Gefitinib (#078-06561, Lot. SKP4099, FUJIFILM Wako) was dissolved in DMSO (#13445-74, Nacalai tesque) at 2 mM and stored at −20 °C as stock solutions. EGF (#059-07873, FUJIFILM Wako) was dissolved in phosphate-buffered saline (PBS) at 100 μg/mL and stored at −20 °C as stock solutions. Streptavidin Sepharose High Performance (#90100484, Cytiva) was stored at 4 °C as stock solutions. Streptavidin-HRP (#ab7403, abcam) (IB 1:10000) was stored at 4 °C as stock solutions. Dynasore (#S8047, Selleck) was dissolved in DMSO (#13445–74, Nacalai tesque) at 80 mM and stored at −20 °C as stock solutions.

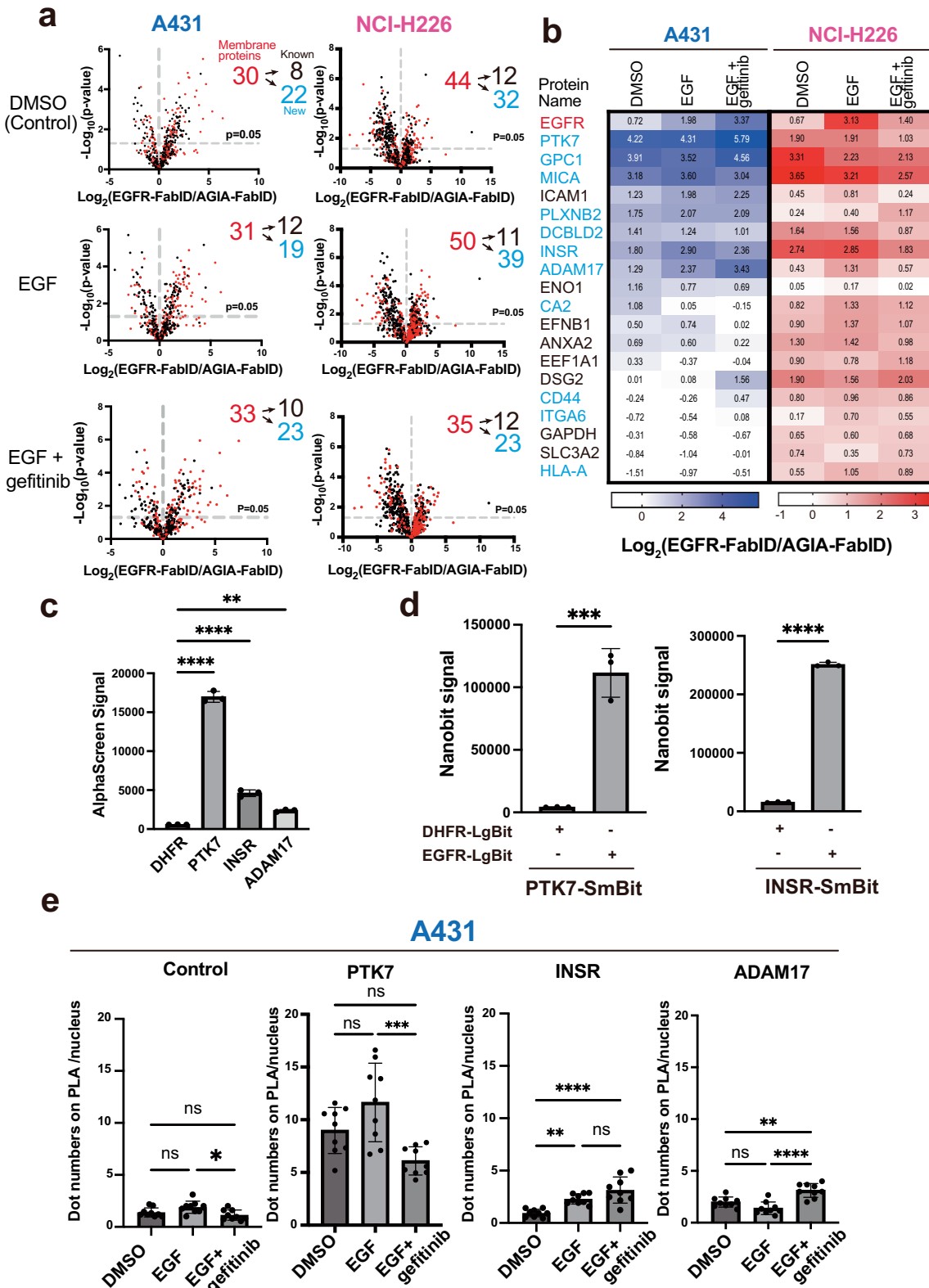

## Plasmids

pcDNA3.1(+) and pcDNA3.4 vectors were purchased from Invitrogen/ Thermo Fisher Scientific and RIKEN, respectively. The pEU vector for wheat cell-free protein synthesis was constructed in our laboratory as previously described[55]. pBiT1.1-C[TK/LgBiT] (# N2014, Promega) and pBiT2.1-C[TK/SmBiT](#N2014, Promega) were purchased from Promega. pcDNA3.4-TEV-His, pEU-bls-MCS, and pEU-FLAG-MCS plasmids were constructed by polymerase chain reaction (PCR) using the In-Fusion system (Takara Bio) or PCR and restriction enzymes. AirID was was constructed in our laboratory, as previously described[24]. *EGFR* and *MDM2* were purchased from the Kazusa Clone collection[56] using pEU-FLAG-GST-EGFR. *EGF, PTK7, OTULIN, ADAM17* and *p53* were purchased from Mammalian Gene Collection (MGC). The *SHB and LHB* genes (S-protein of hepatitis B virus) were kindly provided by Prof. Y. Matsuura

**Fig. 6 | Individual analysis of exPPI response to EGFR in A431 and NCI-H226 cells in the presence of EGF and gefitinib. a** Volcano plot of DMSO (Control) EGF- or EGF+gefitinib-treated zones, each detected as biotinylated peptides by LC-MS/MS. Peptides derived from plasma membrane proteins are indicated by red dots. Numbers in black indicate the number of proteins known to interact with EGFR. Numbers in blue indicate the number of proteins that are unknown interactors with EGFR. Significant changes in the volcano plots and heat map were calculated by Student's two-sided $t$ test and the false discovery rate (FDR)-adjusted $P$-values calculated using Benjamini–Hochberg method are shown in the Supplementary Data 3, 4. **b** Heat map showing exPPI changes upon addition of EGF or gefitinib for cell surface proteins that were commonly biotinylated in A431 and NCI-H226 cells. Each cell was assigned an average value. **c** EGFR interactions were detected using AlphaScreen (independent experiments, $n$ = 3). AlphaScreen with the EGFR extracellular domain and various FLAG fusion proteins. Statistical significance was determined using one-way ANOVA with Dunnett's multiple comparisons test (****$P \le 0.0001$, ***$P \le 0.0002$, **$P \le 0.0021$). Error bars represent standard deviations. **d** Confirmation of live cell interactions using the NanoBiT system. DHFR-LgBit was used as the control. Statistical significance was determined using two sided $t$ test (three biological repecates, ****$P \le 0.0001$, ***$P \le 0.0002$). Error bars represent standard deviations. The cells used were HEK293T cells. **e** Proximity ligation assay (PLA) performed on A431 cells. Bar graph depicting the number of PLA-positive foci compared to each treatment group calculated with Andy's PLA algorithm and quantified using GraphPad Prism 9 software ($n$ = 9 per group, One-way ANOVA with Tukey's post-hoc test, ****$P \le 0.0001$, ***$P \le 0.0002$, **$P \le 0.0021$, *$P \le 0.0332$). Error bars represent standard deviations. Source data are provided as a Source data file.

of the Osaka University, Center for Infectious Diseases Education and Research. The pEU-FLAG-GST-DRD1 CTD was constructed in our laboratory, as previously described[28]. The restriction enzyme sites were added to *EGFR*, *EGFR* extracellular domain, *PTK7*, *OTULIN*, *SHB* and *ADAM17* by PCR. The EGFR extracellular domain was cloned into pEU-MCS and pEU-bls-MCS. EGFR was cloned into pcDNA3.1(+)-MCS and pBiT1.1-C [TK/LgBiT] plasmids. *OTULIN* and *SHB* were cloned into pEU-AGIA-MCS. *PTK7* and *INSR* [28–758aa] were cloned into pEU-FLAG-MCS. *PTK7* and *INSR* was cloned into pBiT2.1 [TK/SmBiT], respectively. ADAM17 and LHBs were cloned into pEU-FLAG-MCS. *MDM2*, and *p53* were cloned into pEU-FLAG-GST or pEU-AGIA vectors using the Gateway cloning system (Thermo Fisher Scientific). cDNAs for the anti-AGIA antibody light chains and EMab-134 light chain were cloned into the pcDNA3.4 expression vector using PCR and In-Fusion Reaction. The anti-AGIA heavy chain Fab fragment or EMab-134 heavy chain Fab fragment was cloned into pcDNA3.4-TEV-His using the In-Fusion HD Cloning Kit (#639649, Takara Bio) together with the AirID or TurboID fragment to generate pcDNA3.4-anti-AGIA Heavy Fab-AirID-TEV-His and pcDNA3.4-EMab-134 Heavy chain-AirID-TEV-His, respectively.

## Cell culture and transfection
Expi293F cells (Gibco/Thermo Fisher Scientific, RRID:CVCL_D615) were shaken at $125 \pm 5$ rpm at $37\,^{\circ}C$ under 8% $CO_2$ in Expi293F medium (#A1435101, Gibco/Thermo Fisher Scientific) supplemented with 100 U/mL penicillin and 100 µg/mL streptomycin (#15140122, Gibco/Thermo Fisher Scientific). A431 (JCRB Cell Bank, RRID:CVCL_0037) cells were cultured in high-glucose DMEM (#043-30085, FUJIFILM Wako) supplemented with 10% foetal bovine serum (FBS; FUJIFILM Wako), 100 U/mL penicillin, and 100 µg/mL streptomycin (#15140122, Gibco/Thermo Fisher Scientific) at $37\,^{\circ}C$ under 5% $CO_2$. HeLa-S3 cells (JCRB Cell Bank, RRID:CVCL_0058) were shaken at $125 \pm 5$ rpm at $37\,^{\circ}C$ under 8% $CO_2$ in Ham's F-12 (#087-08335, FUJIFILM Wako) supplemented with 100 U/mL penicillin and 100 µg/mL streptomycin (#15140122, Gibco/Thermo Fisher Scientific). NCI-H1975 (ATCC, RRID:CVCL_1511) and NCI-H226 (ATCC, RRID:CVCL_1544) were cultured in RPMI160 GlutaMAX medium (#72400047, Gibco/Thermo Fisher Scientific) supplemented with 10% foetal bovine serum (FUJIFILM Wako), 100 U/mL penicillin, and 100 µg/mL streptomycin (#15140122, Gibco/Thermo Fisher Scientific) at $37\,^{\circ}C$ under 5% $CO_2$. Expi293F cells were transiently transfected using Expi Fectamine 293 transfection kit (#A14524, Gibco/Thermo Fisher Scientific).

## Antibodies
The following horseradish peroxidase (HRP)-conjugated antibodies were used in this study: anti-FLAG (#A8592,Sigma-Aldrich, IB 1:5000, RRID:AB_439702), anti-AGIA (produced in our laboratory, IB 1:10000)[28], anti-tubulin (#PM054-7, MBL, IB 1:5000, RRID:AB_10695326), anti-His (#sc-8036, Santa Cruz, IB 1:1000, RRID:AB_627727), and biotin (#7075, Cell Signalling Technology, IB 1:1000, RRID:AB_10696897). The following primary antibodies were used: anti-EGFR (clone EMab-134, IB 1:1000, PLA 1:100, IF 1:100)[34,35],

anti-EGFR (#4267, Cell Signalling Technology, IF 1:100, RRID:AB_2864406), anti-ADAM17 (#3976, Cell Signalling Technology, IF 1:100, PLA 1:100, RRID:AB_2242380), anti-ADAM17 (#sc-390859, SANTA CRUZ, IF 1:100), anti-P-EGFR-Tyr1173 (#4407, Cell Signalling Technology, IB 1:1000, RRID:AB_331795), anti-biotin (#5597, Cell Signalling Technology, IB 1:1000, IF 1:100, RRID:AB_10828011), anti-insulin receptor alpha (#74118, Cell Signalling Technology, PLA 1:100, RRID:AB_2799850), anti-INSR (#A19067, Abclonal, IF 1:100, RRID:AB_2862559), anti-STAT3 (#9132, Cell Signalling Technology, IB 1:1000, RRID:AB_331588), anti-P-STAT3-Y705 (#9145, Cell Signalling Technology, IB 1:1000, RRID:AB_2491009), anti-EEF1A1 (#sc-21758, SANTA CRUZ, IF 1:50, RRID:AB_309663) and anti-PTK7 (#17799-1-AP, proteintech, IF 1:100, PLA 1:100, RRID:AB_2878442). Anti-rabbit IgG (#7074, HRP-conjugated, Cell Signalling Technology, IB 1:10000, RRID:AB_2099233), anti-mouse IgG (#7076, HRP-conjugated, Cell Signalling Technology, IB 1:10000, RRID:AB_330924), F(ab')2-Goat anti-Rabbit IgG (H+L) Cross-Adsorbed Secondary Antibody, Alexa Fluor 555 (#A21431, Thermo Fisher Scientific, IF 1:1000, RRID:AB_2535852), and goat anti-mouse IgG (H+L) Cross-Adsorbed Secondary Antibody, Alexa Fluor™ 488 (#A11001, Thermo Fisher Scientific, IF 1:1000, RRID:AB_2534069) were used as secondary antibodies.

## Preparation of FabID and mAbID
FabID and mAbID were expressed using the Expi293F Expression System (Thermo Fisher Scientific), according to the manufacturer's instructions. The culture medium was purified using protein Ni Sepharose Excel (#GE17371201, Cytiva). The supernatants were added to Ni Sepharose Excel and incubated for 3 h at $4\,^{\circ}C$. The mixture was then washed with three-column volumes of wash buffer (20 mM sodium phosphate, 300 mM NaCl, and 10 mM imidazole). Proteins were eluted in 500 µL fractions with elution buffer (20 mM sodium phosphate, 300 mM NaCl, and 500 mM imidazole). The fractions were dialysed against PBS. Purified FabID and mAbID were frozen and stored at $-80\,^{\circ}C$.

## Immunoblot analysis
Protein samples were separated using SDS-PAGE and transferred onto polyvinylidene difluoride (PVDF) membranes (#IPVH00010, Millipore). The membranes were blocked using 5% skimmed milk (#4273437, Megmilk Snow Brand) in TBST (20 mM Tris-HCl [pH 7.5], 150 mM NaCl, and 0.05% Tween20) at $27\,^{\circ}C$ for 1 h and then treated with the appropriate antibodies. Immobilon (#WBKLS0500 Merck), ImmunoStar LD (#296–69901, FUJIFILM Wako), or EzWestLumi plus (#WSE-7120, Atto) were used as substrates for HRP, and the luminescence signal was detected using an ImageQuant LAS 4000 mini (GE Healthcare). In some blots, the membrane was stripped with a stripping solution (#193-16375, FUJIFILM Wako) and reprobed with other antibodies.

## Wheat cell-free protein synthesis
The recombinant protein was synthesised using a wheat cell-free system. In vitro transcription and wheat cell-free protein synthesis were

performed using the WEPRO1240 Expression Kit (Cell-Free Sciences) or Disulfide Bond PLUS Expression Kit (#CFS-EDX-DB, Cell-free Sciences). Transcription was performed using SP6 RNA polymerase, with plasmids or DNA fragments as templates. The translation reaction was performed in bilayer mode using the WEPRO1240 expression kit (Cell-Free Sciences) or Disulfide Bond PLUS Expression Kit (Cell-Free Sciences), according to the manufacturer's protocol. For biotin labelling of the bls-EGFR extracellular domain, cell-free synthesised crude biotin ligase (BirA) produced using the wheat cell-free expression system was added to the bottom layer, and 0.5 μM (final concentration) of d-biotin (#04822-91, Nacalai Tesque) was added to both the upper and lower layers[8].

### In vitro biotinylation assay with FabID and mAbID

Wheat cell-free expression system synthesised each protein were mixed with FabID or mAbID (Fin 80 ng/μL), added to the reaction mixture, and incubated for 1 h at 26 °C. Next, d-biotin (#04822-91, Nacalai Tesque) was added at a concentration of 500 nM and incubated at 26 °C for 3 h. After the reaction, biotinylated proteins were analysed using SDS-PAGE and immunoblotting.

### Biotinylation of the protein complex by AGIA-FabID

Wheat cell-free expression system synthesised proteins were mixed and incubated for 1 h at 26 °C. In addition, AGIA-FabID (Fin 80 ng/μL) was added to the reaction mixture and incubated for 1 h at 26 °C. Next, d-biotin (#04822-91, Nacalai Tesque) was added at a concentration of 500 nM and incubated at 26 °C for 5 h. After the reaction, biotinylated proteins were analysed using SDS-PAGE and immunoblotting.

### AGIA-FabID pH resistance analysis

One hundred microlitres of protein synthesised using the wheat cell-free protein synthesis system was dialysed in 10 mL of buffer at each pH (100 mM HEPES-NaOH buffer pH = 7.0, 7.2, 7.4, 7.6, and 7.8; 200 mM Tris-HCl buffer pH = 8.0). After 24 h of dialysis, an in vitro biotinylation assay was performed.

### Molecular docking of EGFR-FabID and EGFR-mAbID with EGFR extracellular domains

The complex model of EGFR-FabID and EGFR-mAbID was predicted using AlphaFold2 (AF2)[57]. The predictive antigen-binding region of EGFR-FabID was manually adjusted to dock into epitope regions of the EGFR extracellular domains (Protein Data Bank (PDB) code: 1IVO [https://doi.org/10.2210/pdb1ivo/pdb])). The light and heavy chains of EGFR-FabID were superimposed onto the structure of the anti-canine lymphoma monoclonal antibody (Mab231 and PDB code: 1IGT [https://doi.org/10.2210/pdb1igt/pdb]), and the predictive antigen-binding region of EGFR-mAbID was also manually docked into epitope regions of EGFR extracellular domains (PDB code: 1IVO [https://doi.org/10.2210/pdb1ivo/pdb])). All the molecular structures were generated using PyMOL (Schrödinger).

### Stable cell line

EGFR-overexpressing Expi293F cells were transfected with pcDNA3.1-EGFR in Expi293F cells ($5 \times 10^6$ cells). After 24 h of infection, the culture medium was exchanged, and 1 mg/mL Geneticin G418 (Sigma Aldrich) selection was started 24 h after exchanging the culture medium.

### Cell biotinylation assay with FabID

Biotinylation of EGFR on the plasma membrane was performed using EGFR-overexpressing Expi293F cells transfected with pcDNA3.1-EGFR. Expi293F cells ($5 \times 10^6$ cells) were transfected with pcDNA3.1-EGFR (2 μg) and incubated for 24 h. Next, 385 μL Expi293F serum-free medium (10 mM HEPES) and 115 μL FabID suspension cells reaction buffer [355.5 μg/mL EGFR-FabID or AGIA-FabID, 22.2 mM ATP, 66.7 mM MgCl$_2$, and 333.3 μM d-biotin] were added and incubated at 37 °C for

2 h with rotation. Cells were collected by centrifugation and washed twice with 1 mL of PBS. Cells were lysed with 500 μL RIPA buffer + protease inhibitor (Sigma-Aldrich), followed by sonication and streptavidin pull-down assays.

Expi293F steady EGFR expression cells ($1.0 \times 10^7$ cells) and HeLa S3 cells ($5 \times 10^6$ cells) were resuspended in 385 μL Expi293F medium (serum-free, +10 mM HEPES) or low-glucose D-MEM (#041-29775, FUJIFILM Wako) (serum-free, 10 mM HEPES). Next, 115 μL FabID suspend cell reaction buffer was added and incubated at 37 °C for 2 h with rotation. Cells were collected by centrifugation and washed twice with 1 mL of PBS. Cells were lysed with 500 μL RIPA buffer + protease inhibitor (#P8340, Sigma-Aldrich), followed by sonication and streptavidin pull-down assays.

For A431, NCI-H226, and NCI-H1975 cells, a 10 cm confluent dish was washed once with 2 mL PBS, after which 4.2 mL of high-glucose D-MEM (serum-free, 10 mM HEPES) or RPMI1640 (serum-free, 20 mM HEPES), 800 μL FabID adherent cells reaction buffer (177.8 μg/mL EGFR-FabID or AGIA-FabID, 22.2 mM ATP, 66.7 mM MgCl$_2$, and 333.3 μM d-biotin) were added and incubated at 37 °C for 2 or 6 h. After the reaction, the cells were washed with PBS (2.5 mL) and dissolved in 500 μL of RIPA buffer + protease inhibitor (#P8340, Sigma-Aldrich). After sonication, streptavidin pull-down assays were performed.

### Streptavidin pull-down assay (STA-PDA) using suspension or adherent cells

Streptavidin Sepharose (20 μL/L sample) was washed three times in wash buffer (50 mM Tris-HCl pH 7.5, 100 mM NaCl, 1% SDS), resuspended in 200 μL of wash buffer, and added to the sample. The mixture was then rotated at room temperature for 1 h, and the supernatant was removed by centrifugation. Streptavidin Sepharose was washed three times with 1 mL of wash buffer and boiled in 2× sample buffer (40 μL) at 100 °C for 10 min. The supernatant was recovered by centrifugation.

### Kinase assay

A431 ($3.0 \times 10^5$ cells/mL) cells were seeded in 24-well plates. After 24 h, the cells media was replaced with serum-free media, and cells were treated with DMSO (final 0.1%) or 2 μM gefitinib for 2 h. Next, EGF (100 ng/mL) was added for 5 min, and cells were treated with 80 ng/μL EGFR-FabID. After 15 min, the cells were collected and lysed in 100 μL of 2× sample buffer + 10% mercaptoethanol. Phosphorylation of EGFR and STAT3 activated by EGF stimulation was confirmed by immunoblotting using specific antibodies.

### Flow cytometry

Expi293F cells stably expressing EGFR were biotinylated using EGFR-FabID for 2 h as described in this section above (Cell biotinylation assay with FabID). The cells without EGFR-FabID or Biotin were added PBS instead. After biotinylation of cells by EGFR-FabID, cells were washed three times with 1 mL of PBS. The cells were fixed with 4% paraformaldehyde (#163–20145; FUJIFILM Wako). The cells were then stained with Alexa Fluor 488-conjugated Streptavidin (#S32354, Thermo Fisher Scientific, 1:100, RRID: AB_2315383) suspended in PBS with 5% BSA for 1 h at 4 °C. The stained cells were washed three times with 1 mL of PBS and were replaced with FACS buffer (PBS with 2% FBS, 2 mM EDTA and 0.01% NaN$_3$). Flow cytometry was performed using BD FACSLyric™ Flow Cytometer (BD Biosciences) and the results were analyzed using FlowJo software (Tree Star, Ashland, OR, USA). The filters used for BD FACSLyric™ Flow Cytometer were FITC (excitation wavelength 488 nm, emission wavelength 530/30 nm).

### MTS assay

A431 ($3.0 \times 10^5$ cells/mL) cells were seeded in 96-well plates. After 24 h, the cell medium were replaced with 80 μL serum-free media, and the cells were treated with 20 μL FabID adherent cell reaction buffer

(177.8 μg/mL EGFR-FabID, AGIA-FabID, or PBS: 22.2 mM ATP, 66.7 mM MgCl$_2$, and 333.3 μM d-biotin). After 2 h, cell viability was measured using the CellTiter 96® AQueous One Solution Cell Proliferation Assay (#G3582, Promega), according to the manufacturer's protocol. The absorbance at 490 nm was measured using SpectraMax™ iD3 Multimode microplate reader (Molecular Devices).

## Biotinylation of EGFR by EGFR-FabID during inhibition of endocytosis

NCI-H226 cells in 6 cm confluent dish was washed once with 2 mL PBS, and 2.1 mL of RPMI1640 (serum-free, 20 mM HEPES) were added followed by treatment with DMSO or 50 μM Dynasore (#S8047, Selleck) for 1 h. Next, 400 μL FabID adherent cell reaction buffer was added to the medium and incubated at 37 °C for 3 h. After the reaction, the cells were washed three times with 1 mL of PBS and dissolved in 500 μL of RIPA buffer + protease inhibitor (#P8340, Sigma-Aldrich). Biotinylated cell lysates were analysed by SDS-PAGE and immunoblotting.

## Preparation of cell lysates treated with EGFR-FabID for the enrichment of biotinylated peptides

EGFR biotinylation on the plasma membrane of pcDNA3.1-EGFR-transfected EGFR-overexpressing, Expi293F cells was performed using EGFR-FabID in three independent wells (three biological replicates; $n = 3$). Expi293F cells ($5 \times 10^6$ cells) transfected with pcDNA3.1-EGFR for 24 h. Next, 385 μL Expi293F medium (10 mM HEPES) and 115 μL FabID suspend cell reaction buffer were added and incubated at 37 °C for 2 h with rotation. Cells were collected by centrifugation and washed twice with 1 mL HEPES-Saline (20 mM HEPES-NaOH, pH 7.5, 137 mM NaCl). The cells were then lysed in 500 μL Gdm-TCEP buffer (6 M guanidine-HCl, 100 mM HEPES-NaOH pH 7.5, 10 mM TCEP, and 40 mM chloroacetamide). After lysing the cells, three independent lysates were pooled in one tube and divided into three tubes of 500 μL each. Each tube was subjected to enrichment of biotinylated peptides followed by mass spectrometry.

Biotinylation of EGFR on plasma membrane using Expi293F steady EGFR expression cells was performed with EGFR-FabID in three independent wells (three biological replicates; $n = 3$). Expi293F steady EGFR expression cells ($1.0 \times 10^7$ cells) were resuspended in 385 μL Expi293F medium (serum-free, +10 mM HEPES). Next, 115 μL FabID suspend cell reaction buffer was added and incubated at 37 °C for 2 h with rotation. Cells were collected by centrifugation and washed twice with 1 mL HEPES-Saline (20 mM HEPES-NaOH, pH 7.5, 137 mM NaCl). The cells were then lysed in 500 μL Gdm-TCEP buffer (6 M guanidine-HCl, 100 mM HEPES-NaOH pH 7.5, 10 mM TCEP, and 40 mM chloroacetamide). After lysing the cells, three independent lysates were pooled in one tube and divided into three tubes of 500 μL each. Each tube was analysed by mass spectrometry.

Biotinylation of A431 cells by EGFR-FabID was performed in three independent 10 cm dishes. A 10 cm dish confluent with A431 cells in three biological replicates was washed once with 1 mL of PBS. Then, 2.1 mL of DMEM (serum-free, 10 mM HEPES) was added, and cells were treated with DMSO or 2 μM gefitinib (DMSO final 0.1%) for 2 h, followed by treatment with 100 ng/mL EGF. Next, 800 μL FabID adherent cell reaction buffer was added to the medium and incubated at 37 °C for 2 h. After the reaction, cells were washed with 5 mL Hepes-Saline (20 mM HEPES-NaOH, pH 7.5, 137 mM NaCl) and lysed in 500 μL Gdm-TCEP buffer (6 M guanidine-HCl, 100 mM HEPES-NaOH pH 7.5, 10 mM TCEP, and 40 mM chloroacetamide). Cell lysates were pooled in one tube and divided into three tubes of 500 μL each. Each tube was subjected to enrichment of biotinylated peptides followed by mass spectrometry.

Biotinylation of NCI-H226 cells by EGFR-FabID was performed in five independent 10 cm dishes. Confluent NCI-H226 cells (10 cm dish, five biological replicates) were washed once with 1 mL PBS. Then,

2.1 mL of RPMI1640 (serum-free, 20 mM HEPES) was added, and cells were treated with DMSO or 2 μM gefitinib (DMSO final 0.1%) for 1 h, followed by treatment with 5 μL EGF (100 μg/μL). Next, 800 μL FabID adherent cell reaction buffer was added to the medium and incubated at 37 °C for 3 h. After the reaction, cells were washed with 5 mL Hepes-Saline (20 mM HEPES-NaOH, pH 7.5, 137 mM NaCl) and lysed in 300 μL Gdm-TCEP buffer (6 M guanidine-HCl, 100 mM HEPES-NaOH pH 7.5, 10 mM TCEP, and 40 mM chloroacetamide). Cell lysates were pooled in one tube and divided into three tubes of 500 μL each. Each tube was subjected to enrichment of biotinylated peptides followed by mass spectrometry.

## Enrichment of biotinylated peptides using Tamavidin 2-REV

All experiments were performed in triplicate for each treatment. The cell lysates in Gdm-TCEP buffer were dissolved by heating and sonication and then centrifuged at 20,000 × $g$ for 15 min at 4 °C. The supernatants were recovered and proteins were purified by methanol–chloroform precipitation and solubilised using PTS buffer (12 mM SDC, 12 mM SLS, 100 mM Tris-HCl, pH8.0). After sonication and heating, the protein solution was diluted 5-fold with 100 mM Tris-HCl, pH8.0 and digested with trypsin (MS grade, Thermo Fisher Scientific) at 37 °C overnight. The resulting peptide solutions were diluted 5-fold with TBS (50 mM Tris-HCl, pH 7.5, 150 mM NaCl). Biotinylated peptides were captured on a 15 μL slurry of MagCapture HP Tamavidin 2-REV magnetic beads (#133-18611, FUJIFILM Wako) after incubation for 3 h at 4 °C. After washing with TBS five times, the biotinylated peptides were eluted with 100 μL of 1 mM biotin in TBS for 15 min at 37 °C twice. The combined eluates were desalted using GL-Tip SDB (#7820-11200, GL Sciences), evaporated in a SpeedVac concentrator (Thermo Fisher Scientific), and re-dissolved in 0.1% TFA and 3% acetonitrile (ACN).

## Data-dependent LC-MS/MS analysis

LC-MS/MS analysis of the resultant peptides was performed on an EASY-nLC 1200 UHPLC connected to an Orbitrap Fusion mass spectrometer using a nanoelectrospray ion source (Thermo Fisher Scientific). The peptides were separated on a 150-mm C$_{18}$ reversed-phase column with an inner diameter of 75 μm (Nikkyo Technos) using a linear 4–32% ACN gradient for 0–60 min, followed by an increase to 80% ACN for 10 min. The mass spectrometer was operated in data-dependent acquisition mode with a maximum duty cycle of 3 s. The MS1 spectra were measured with a resolution of 120,000, an automatic gain control (AGC) target of $4 \times 10^5$, and a mass range of 375–1500 $m/z$. HCD MS/MS spectra were acquired in a linear ion trap with an AGC target of $1 \times 10^4$, an isolation window of 1.6 $m/z$, a maximum injection time of 200 ms, and a normalised collision energy of 30. Dynamic exclusion was set to 10 s. Raw data were directly analysed against the Swiss-Prot database restricted to *Homo sapiens* using Proteome Discoverer version 2.4 (Thermo Fisher Scientific) with the Sequest HT search engine. The search parameters were as follows: (a) trypsin as an enzyme with up to two missed cleavages, (b) precursor mass tolerance of 10 ppm, (c) fragment mass tolerance of 0.6 Da; (d) carbamidomethylation of cysteine as a fixed modification, and (e) acetylation of protein N-terminus, oxidation of methionine, and biotinylation of lysine as variable modifications. Peptides were filtered at a false discovery rate (FDR) of 1% using the Percolator node. Label-free quantification was performed based on the intensities of the precursor ions using a precursor ion quantifier node. Normalisation was performed such that the total sum of the abundance values for each sample over all peptides was the same. For statistical analyses of the MS data, the *P*-values in each volcano plot were calculated using Student's *t* tests by using Microsoft Excel (version 16.78.3). The adjusted *P*-values were calculated by controlling the FDR by using Microsoft Excel (version 16.78.3) and are shown in the Supplementary Data. Proteins with

statistically significant modification levels were identified using the Student's $t$ test ($P < 0.05$) between EGFR-FabID triplicates and AGIA-FabID control triplicates. Additionally, a protein was determined enriched only when it showed at least a 1-fold change over the AGIA-FabID control.

## GST pull-down assay

pcDNA3.4-FLAG-GST-TEV-His-AGIA-GS linker-EGF (FLAG-GST-EGF) was constructed in the form of a TEV-His-AGIA-GS linker inserted between FLAG-GST and EGF. The secreted signal (MGILPSPGMPALLSLVSLLSVLLLMGCVAETG)[58] was fused before FLAG-GST. The FLAG-GST-EGF plasmid was transfected into Expi293F cells stably expressing EGFR ($5 \times 10^6$ cells) in 2 mL Expi293F medium. After 24 h of transfection, enhancer of Expi293F Transfection Kit (#A14525, Thermo Fisher Scientific) was added. After 24 h of enhancer addition, 200 μL EGFR-FabID or PBS and 200 μL biotin mix buffer (5 mM HEPES, 500 μM ATP, 5 μM $MgCl_2$ and 50 μM d-Biotin) or PBS were added and incubated at 37 °C for 2 h with rotation. Cells and cell culture medium were collected by centrifugation. Supernatant (40 μL) was collected as sup section. Then NaCl (final 150 mM) and DTT (final 10 mM) are added to the culture medium. A 30 μL slurry of Glutathione Sepharose™ 4B (#17-0756-01, Cytiva) washed with PBS and suspended in 100 μL PBS is added to the culture medium. The beads were washed 3 times with 1 mL PBS after 3 h rotation at 27 °C. After boiling with 2×sample buffer (40 μL), biotinylation of eluted proteins was analyzed by immunoblotting.

## AlphaScreen-based biochemical assays using recombinant proteins

The bls-EGFR extracellular domain and FLAG fusion proteins were synthesised using a Disulfide Bond PLUS Expression Kit (#CFS-EDX-DB, CellFree Science). Proteins synthesised using the wheat cell-free expression system were mixed in 15 μL of reaction buffer (100 mM Tris-HCl pH 8.0, 1 mg/mL BSA, 100 mM NaCl, and 0.1% Tween20) and incubated at 26 °C for 1 h. Then, 10 μL of the detection mixture [1 mg/mL BSA, 100 mM NaCl, and 0.1% Tween20, 0.04 mg/mL protein A acceptor beads (PerkinElmer) and 0.04 mg/mL streptavidin donor beads (PerkinElmer)] in the reaction buffer were mixed. After incubation at 26 °C for 1 h, the chemiluminescence signal was detected using an Envision microplate reader (PerkinElmer). The Envision microplate reader filter detected light between 520 and 620 nm at an excitation wavelength of 680 nm.

## Immunofluorescent staining

A431 cells were cultured on 13 mm poly L-lysine-coated glass slides (#C1110, Matsunami) in 24-well plates. After culturing A431 for 24 h, 24-well plates were washed once with 200 μL PBS, after which 380 μL of high-glucose D-MEM (serum-free, 10 mM HEPES) and 120 μL FabID adherent cell reaction buffer (177.8 μg/mL EGFR-FabID, 22.2 mM ATP, 66.7 mM $MgCl_2$, and 333.3 μM d-biotin) were added to the A431 cells and incubated at 37 °C for 2 h. After one wash with 500 μL PBS, A431 cells were fixed with 4% paraformaldehyde in PBS for 15 min at room temperature. Cells were incubated with anti-EGFR (clone EMab-134[34,35]), anti-biotin, and 4′,6-diamidino-2-phenylindole (DAPI) overnight at 4 °C after blocking with 0.5% CS in TBST overnight. After washing with TBST at 27 °C for 15 min, cells were incubated in F(ab')2-Goat anti-Rabbit IgG (H + L) Cross-Adsorbed Secondary Antibody, Alexa Fluor™ 555 and Goat anti-Mouse IgG (H + L) Cross-Adsorbed Secondary Antibody, and Alexa Fluor™ 488 at 27 °C for 1 h. After washing with TBST at room temperature for 15 min, the stained cells were mounted using ProLong™ Gold Antifade Mountant with DAPI (#P36935, Invitrogen/Thermo Fisher Scientific) and observed under a BZ-X810 Microscope (Keyence, Osaka, Japan).

A431 and NCI-H226 cells were cultured on 13 mm poly L-lysine-coated glass slides (#C1110, Matsunami) in 24-well plates. After 24 h of culture, cells were washed twice with PBS. Cells were fixed with 4% paraformaldehyde in PBS for 15 min at room temperature and permeabilised with 0.01% digitonin in PBS at 27 °C for 15 min. For the EEF1A1 treatment section, no permeability treatment was performed. Then, the cells were incubated with 0.5% cattle serum (CS) in TBST at 4 °C for 1 h. A431 cells were incubated with anti-ADAM17 (#sc-390859, SANTA CRUZ, 1:100), anti-EGFR (Cell Signalling Technology, #4267, 1:100), or anti-PTK7 (Proteintech, 17799-1-AP, 1:100), and anti-EGFR (clone EMab-134[34,35], 1:100) antibodies. NCI-H226 cells were incubated with anti-INSR (#A19067, Abclonal, 1:100) or anti-EEF1A1 (#sc-21758, SANTA CRUZ, 1:50) and anti-EGFR (clone EMab-134; produced in our laboratory, 1:100) antibodies or anti-EGFR (Cell Signalling Technology, #4267, 1:100). After washing with TBST at 27 °C for 15 min, the cells were incubated in DAPI, F(ab')2-Goat anti-Rabbit IgG (H + L) Cross-Adsorbed Secondary Antibody, Alexa Fluor™ 555 and Goat anti-Mouse IgG (H + L) Cross-Adsorbed Secondary Antibody, and Alexa Fluor™ 488 at room temperature for 1 h. After washing with TBST at room temperature for 15 min, the stained cells were mounted using ProLong™ Gold Antifade Mountant with DAPI (#P36935, Invitrogen/Thermo Fisher Scientific) and observed under a BZ-X810 Microscope (Keyence, Osaka, Japan). The filters used for the BZ-X810 Microscope were BZ-X Filter GFP (excitation wavelength 470/40 nm, emission wavelength 525/50 nm), BZ-X Filter TRITC (excitation wavelength 545/25 nm, emission wavelength 605/70 nm), and BZ-X Filter DAPI (excitation wavelength 360/40 nm, emission wavelength 460/50 nm) (Keyence).

## NanoBiT PPI assay

Plasmids (25 ng of pBiT2.1-PTK7-SmBiT or pBiT2.1-INSR-SmBiT together with 25 ng of pBiT1.1-DHFR-LgBiT or pBiT1.1-EGFR-LgBiT) and 50 ng of pcDNA3.1-(+) vectors were transfected into HEK293T cells in 96 well-plate using polyethyleneimine (PEI) Max (MW 40,000) (Poly-Science, Inc.) After 24 h, the culture medium was replaced with 80 μL Opti-MEM reduced serum medium (Gibco). Then, 20 μL Nano-Glo Live Cell Reagent (Promega) was added and measured NanoBiT signal using the GloMAX Discover System (Promega). Three wells of cells cultured and transfected in independent wells were tested. (three biological replicates; $n = 3$)

## Proximity ligation assay (PLA)

A431 cells were cultured on 13 mm poly L-lysine-coated glass slides (#C1110, Matsunami) in 24-well plates. After culturing for 24 h, the cells were washed two times with PBS. Cells were fixed with 4% paraformaldehyde in PBS for 15 min at room temperature and permeabilised with 0.1% Triton X-100 in PBS (PTK7 and ADAM17); INSR was not amenable to permeabilisation. The cells were then washed twice with PBS and blocked with Duolink block solution for 1 h at 37 °C. Next, the cells were stained overnight at 4 °C with primary antibodies diluted in Duolink antibody dilution buffer. Then, the cells were washed twice with wash buffer A (10 mM Tris, pH-7.4, 150 mM NaCl, and 0.05% Tween) and incubated with 20 μL of secondary antibody [4 μL anti-mouse PLUS antibody (#DUO82001, Sigma-Aldrich) + 4 μL anti-rabbit MINUS antibody(#DUO82005, Sigma-Aldrich) + 12 μL Duolink® dilution buffer] at 37 °C for 1 h. The secondary antibody mix was gently aspirated and washed twice with wash buffer A. Fifteen microliters of ligation mix was added to each sample and incubated at 37 °C for 30 min. Slides were washed twice with wash buffer A for 2 min each. Fifteen microliters of the polymerase mix was added to each sample and incubated for 100 min at 37 °C. The amplification mix was aspirated and washed twice with wash buffer B (200 mM Tris, pH-7.5, 100 mM NaCl) for 10 min each. The slides were washed once with 0.01× buffer B for 1 min. After aspirating buffer B, slides were then mounted using Duolink® in situ mounting medium with DAPI (#DUO82040, Sigma-Aldrich) and observed under a BZ-X810 Microscope (Keyence).

The filters used for the BZ-X810 Microscope were BZ-X Filter GFP (excitation wavelength 470/40 nm, emission wavelength 525/50 nm) and BZ-X Filter DAPI (excitation wavelength 360/40 nm, emission wavelength 460/50 nm) (Keyence). PLA foci and the number of nuclei were measured using Andy's PLA algorithm[59]. The number of PLA foci was measured in nine separate photographs using Andy's PLA algorithm (biological replicates, $n = 9$). A table of the optimised image parameters used for the PLA image analysis is provided in the source data.

## Bioinformatics analyses

Proteins predominantly biotinylated were identified using EGFR-FabID. To determine the subcellular localisation of the biotinylated proteins, the TMHMM (https://services.healthtech.dtu.dk/service.php?TMHMM-2.0) software and the UniProt database were employed. If a protein has transmembrane helices or is annotated as membrane localised, it was considered as a membrane located protein. If the biotinylation site of the membrane located protein was predicted as extracellular region, it was considered to be located on extracellular region. The known protein-protein interaction data were downloaded from the IntAct database (https://www.ebi.ac.uk/intact). Pathway diagrams of protein-protein interaction network based on data analysed by DTX (https://harrier.nagahama-i-bio.ac.jp/dtx/) were generated using the Cytoscape software (https://cytoscape.org).

## Statistical analyses

Significant changes were analysed by Student's $t$ tests using Microsoft Excel spreadsheets with a basic statistical programme or one-way ANOVA followed by Tukey's post-hoc test using GraphPad Prism 9 software (GraphPad, Inc.). Significant changes in the volcano plots were calculated by Student's two-sided $t$ test and the false discovery rate (FDR)-adjusted $P$-values calculated using Benjamini–Hochberg method by Microsoft Excel (Microsoft) are shown in the Supplementary Data. For all tests, $P < 0.05$ was considered statistically significant. Immunoblot analyses and streptavidin pull-down assays were repeated more than twice, with similar results.

## Reporting summary

Further information on research design is available in the Nature Portfolio Reporting Summary linked to this article.

## Data availability

The MS proteomics data have been provided in Supplementary Data 1–8 and deposited to the ProteomeXchange Consortium via the jPOST partner repository with the dataset identifiers PXD039449 (Proximity biotinylation of Expi293F cells overexpressing EGFR by EGFR-FabID), PXD043525 (Proximity biotinylation of Expi293F cells stably expressing EGFR by EGFR-FabID), PXD039450 (Proximity biotinylation of A431 cells by EGFR-FabID), PXD039451 (Proximity biotinylation of NCI-H226 cells by EGFR-FabID), PXD043526 (Proximity biotinylation of A431 cells in the gefitinib alone treatment by EGFR-FabID), and PXD043527 (Proximity biotinylation of NCI-H226 cells in the gefitinib alone treatment by EGFR-FabID). Source data are provided with this paper.

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

## Acknowledgements

The authors thank for Y. Adachi technical assistance and the Applied Protein Research Laboratory of Ehime University. We thank Prof. Y. Matsuura (Osaka University) for providing *LHB* and *SHB* genes. We would like to thank the Applied Protein Research Laboratory of Ehime University. This work was partially supported by the Platform Project for Supporting Drug Discovery and Life Science Research [Basis for Supporting Innovative Drug Discovery and Life Science Research (BINDS)] from the Japan Agency for Medical Research and Development (AMED) under Grant Number JP21am0101077 (T.Sawasaki.), 22ama121010j0001 (T.S.), 23ama121010j0002 (T.S.), JP22ama121008 (Y.K.). This work was also partially supported by JSPS KAKENHI (19H03218 and JP21K19230 for T.S.), Joint Usage and Joint Research Programs of the Institute of Advanced Medical Sciences, Tokushima University (H.K., T.S.), and Takeda Science Foundation. We would like to thank Editage (www.editage.com) for English language editing.

## Author contributions

K.Y. performed cloning and characterisation of AGIA-mAbID, EGFR-mAbID and EGFR-FabID, cell-based assays, and biotinylation assay; R.S. performed cloning and analysis of AGIA-FabID; K.N. and H.K. performed enrichment of biotinylated peptides and LC-MS/MS analyses; H.F. performed structural modelling; M.K.K. and Y.K. cloned antibodies; A.H. and T.Shirai. performed bioinformatic analyses of biotinylated sites and proteins; K.Y. and T.Sawasaki. analysed the data and wrote the draught paper; T.Sawasaki. conceived the research and designed the study; H.K. and T.Sawasaki. designed the experiments, wrote the paper, and all authors revised the manuscript.

## Competing interests

The authors declare no competing interests.
