## [Peer Review File · Nature Communications]

Proximity extracellular protein-protein interaction analysis of EGFR using AirID-conjugated fragment of antigen bindingReviewer #1 (Remarks to the Author):

Review Kohdai Yamada et. Al.

The authors present a new method based on proximity biotinylation (AirID) to identify extracellular EGFR interactors by LC-MS/MS analysis. With this EGFR-FabID-AirID fusion construct they identified several proteins after EGF stimulation and EGFR inhibitor treatment that are in proximity of EGFR and potentially interact with EGFR in the extracellular space. This is a potentially interesting method and approach to screen for novel extracellular interacting complexes. While my expertise is not sufficient to comment on technical details of the robustness of the methodology, I have several conceptual concerns related also to the physiological relevance of some of the potential EGFR interactors.

1) Have the authors shown that the EGFR-FabID/EGFR-mabID do not internalize after binding to EGFR?

2) With the Expi293F cell line grown in suspension and transiently overexpressing EGFR the authors find that also intracellular sites of the EGFR are biotinylated. How can this happen? Is this due to internalization of the complex when bound to EGFR? The explanation provided on page 16 lines 284-285 seems a bit weak...

3) Related to 2), why are intracellular sites not biotinylated in cells growing in adherence?

4) In all the biotinylation experiments with Gefitinib, the control with Gefitinb alone is missing. It would be important to show whether extracellular EGFR interactions are affected in catalytically inactive EGFR in the absence of ligand.

5) Among the interactors found there is EEF1A1, a translation elongation factor and an intracellular protein. It is extremely surprising that it is found outside the cell in the extracellular space and to interact with EGFR.... Is EEF1A1 leaked out from dying cells or how do the authors explain these results? Is this a specific and relevant interaction and what is the physiological function should this have?

6) All the results shown suggest interactions with EGFR which are however not demonstrated to occur in live cells. Moreover, proximity is no proof of interaction. Without a direct biological proof, the results could simply be an artifact. Some interactions with truncated proteins (extracellular portion) are shown in Fig. 6c, but to exclude artifacts, experiments in a more physiological setting should be performed, e.g. IF, live imaging, etc., demonstrating also the physiological relevance.

7) It is surprising that EGFR ligands themselves were not biotinylated in these assays. These would have been the perfect positive control for the approach.

Minor:

- Was the experiment shown in Fig 3 with Expi293F cells done in the presence or absence of EGF? This is not clear from the text and in Fig3a the scheme contains also EGF.

- The author could indicate the sites recognized by EGF making it more evident that EGF and EGFR-FabID do not interfere when given together.

- the English should be improved

Reviewer #2 (Remarks to the Author):

The manuscript by Yamada et al. reports the develop a method to uncover extracellular protein-

protein interactions. For this they use the AirID proximity labeling enzyme their group previously developed. Their approach fuses AirID to either full length IgG antibody (mAbID) or fragments of antigen binding (FabID). FabID outperformed mAbID in tests using anti-AGIA mAb or EGFR mAb. They go on to nicely show that EGFR-FabID can identify extracellular interactors of EGFR. interestingly, they also show that this approach can identify proteins that interact with EGFR dependent on treatment with EGF ligand or and EGFR inhibitor (gefitinib).

- Overall, the manuscript is nicely written and technically sound. Though the use of AirID, mAbID, and FabID can get a bit confusing.

- This approach relies on having well characterized monoclonal antibodies for the protein of interest, which limits how widely it can be adopted. It is mentioned in the introduction that previous BioID based methods to identify extracellular interactors used genetic insertions. Perhaps the authors can expand on how the monoclonal antibody approach is needed or improves direct fusions to the protein of interest.

- The authors find a number of enriched EGFR interactors it would be nice to show if any are biologically meaningful, but perhaps this is beyond the scope of this manuscript.

Reviewer #3 (Remarks to the Author):

Proximity extracellular protein-1 protein interaction analysis of EGFR using AirID-conjugated fragment of antigen binding

Overall Comments:

This manuscript describes a novel and sensitive approach to detect proximity between the extracellular domains of proteins from both adherent and suspension cell lines. The epidermal growth factor receptor was used to develop the method, and the results were validated in multiple cell lines and several key interactions that were previously unknown were confirmed by orthogonal detection methods. By linking biotinylation events to specific peptides, the mass-based readout facilitates structural interpretations. Additionally, the method was demonstrated to quantify changes in multiprotein complexes that result from treatments with a ligand and small molecule. Thus, this manuscript introduces a valuable new method to the biomedical research community that could be applied to interrogate many complex physiological systems. Overall this manuscript is mostly well-written and clear to understand; however, I would recommend that the authors carefully review the text for errors and make additional edits to the grammar to improve the clarity. In many places throughout the manuscript and figure legends it is stated "three biological replicates" but it is not clear what is being shown in the figures. Are the data shown (1) from a single experiment that is "representative" of three biological replicates or (2) the mean of three biological replicates? Please clarify this throughout the manuscript and show the mean +/- standard deviation or standard error of the mean (as appropriate) wherever replicate data exists. Also, there are a number of discrepancies between the figures and their legends. Please review these carefully and ensure that the legend corresponds to the figures as shown. Also, please make sure that all acronyms and abbreviations are defined.

Specific Comments about the Main Text:

1. Page 7, lines 116 – 117: "Biotinylation of these purified proteins was mainly found in the heavy chain but not in each light chain (Fig. 1c)." Please mark the positions of the light and heavy chains in Figure 1(c).
2. Page 10, line 169, "To understand why FabID was more highly biotinylated on cells than mAbID, ...". I think the authors meant to state, "To understand why the EGFR is more highly biotinylated on cells by EGFR-FabID than EGFR-mAbID"?
3. Page 10, lines 176 - 177: "The productivity of TurboID-fusion Fab was lower than AirID-based

EGFR-FabID in Expi293F cells (IB: His in Supplementary Fig. 1b), ..." Should the word "productivity" be replaced with "expression" based on the anti-His immunoblot, which shows a greater abundance of EGFR-FabID?

4. Pages 11 – 12, lines 187 – 199: "MS analysis indicated that EGFR-FabID biotinylated five and seven lysine residues in the extracellular and intracellular regions of EGFR, respectively (Fig. 2f)". It is not clear to me how the EGFR-FabID was able to biotinylate the seven intracellular lysine residues. Is it thought that this occurred after cell lysis? Also, what distances are the residues K716 – K1188 from Ser380? Is it possible that, following cell lysis, biotinylation occurs "in trans" among EGFR proteins and independent of proximity among both intracellular and extracellular regions of proteins? If so, that will impact the interpretation of extracellular peptides that are found to be biotinylated. If my interpretations are valid, I wonder if it is possible or if the authors considered to quench the enzyme prior to cell lysis?

5. Page 13, line 222: Why are the 10 new proteins being referred to as "clones"?

6. Page 13, lines 227 – 228: I am not sure what is meant by "mediated by one another protein". Please clarify this statement.

7. Page 13, line 229: "...proximal exPPIs induced by EGFR-FabID..." I don't think that EGFR-FabID is supposed to be inducing interactions, it is providing a way to detect them, right?

8. Page 21, line 377: Proximity is misspelled as "ploximity".

9. Page 21, lines 377 – 378: "As shown in Fig. 6b, proximity exPPI of PTK7 with EGFR was reduced by gefitinib treatment." In Figure 6b based on the data from A431 cells, it looks like biotinylation (i.e. proximity) to PTK7 is increased with gefitinib treatment, not reduced; however, it appears that biotinylation (i.e. proximity) to PTK7 is reduced with gefitinib treatment in NCI-H226 cells. Please clarify this statement in the text. Additionally, it is not clear in Figure 6 (e) or the figure legend that the data are from A431 cells.

Specific Comments about the Materials and Methods Section:

10. Please comment about how all of the cell lines were authenticated.

11. All reagents, supplies, kits, and materials listed must include the manufacturer and catalog number, as well as lot numbers wherever possible. In addition, Research Resource Identifiers (RRIDs) (<https://scicrunch.org/resources>) must also be included wherever possible to clarify exactly which materials, supplies, and reagents were used. The RRIDs are particularly important to specify the exact antibodies and cell lines used. All of this information is critical to enable others to leverage this valuable new proximity method being described and to improve the reproducibility of biomedical research in general.

12. Page 28, line 556: Each is misspelled as "eath".

13. Page 34, lines 700 – 703: Only volumes are given for the two types of AlphaLISA beads, but the final concentrations are what is important. Also the filters used for this assay in the Envision microplate reader should be described. Finally, it might be more appropriate to refer to a "chemiluminescent signal" than a "luminescence signal".

14. Page 35, lines 712 – 713: Please indicate the BZ-X810 microscope settings, including excitation/emission filters and excitation wavelength(s).

Specific Comments about the Figures:

15. Figure 1. In panel (c) detection of AGIA-FabID and AGIA-mAbID by anti-polyhistidine is not mentioned in the figure legend. What is FG-DRD1-C? Is it FLAG-GST-DRD1? Please clarify this in the legend. In panel (c) please mark the positions of the light and heavy chains. In panel (h), please show the position of FLAG-p53 in the Western IB: Biotin blot.

16. Figure 3. In panel (a) should the 10 new molecules identified be referred to as "proteins" rather than "clones"? I do not see panel (e) in the figure provided. I also do not see any data for Alphascreens with the EGFR extracellular domain and various FLAG fusion proteins. It appears that (d) shows the pathway analysis of extracellular proteins detected using mass spectrometry. Please revise this figure and associated legend.

17. Figure 4. In panel (c) data is shown for A431, but not NCI-H226 cells, whereas the legend indicates data are shown for both. In panel (d) "transiently" is misspelled. The data in panel (e) should not be shown as a heatmap – the data should be graphed as a scatterplot with all replicate data points shown.

18. Figure 5. In panel (b) "transiently" is misspelled (in green).

19. Figure 6. In panel (b) this data should not be shown as a heatmap – the data should be graphed as a scatterplot with the three replicate data points shown with error bars (as in panels c - e). Doing this might require the data for A431 and NCI-H226 to be shown in separate graphs/panels. In panels c - e, the data show replicates and error bars. Do the error bars indicate standard deviation or standard error of the mean? Please define "PLA" as "proximity ligation assay". Finally it is not clear in panel (e) or the figure legend that the data are from A431 cells. Please clarify this in both the figure and in the legend.

20. Supplementary Fig. 2: Were the data shown in Tables a - c (by the way, "c" is not labeled) from single experiments or were replicates performed? Please clarify that in the figure legend. If replicates were performed, please indicate the type (independent or technical replicates) and include all the data (mean +/- standard deviation) within the tables.

Point-by-Point Responses to the Reviewers' Critiques

We deeply appreciate the thorough analysis and constructive suggestions provided by the three reviewers to further improve our manuscript. As described in more detail below, we have addressed all the reviewers' concerns. With this revision, we hope that the reviewers will concur with us that we have addressed all the raised concerns in a satisfactory manner and, consequently, substantially strengthened our paper.

Reviewer #1 (Remarks to the Author):

Review Kohdai Yamada et. Al.

The authors present a new method based on proximity biotinylation (AirID) to identify extracellular EGFR interactors by LC-MS/MS analysis. With this EGFR-FabID-AirID fusion construct they identified several proteins after EGF stimulation and EGFR inhibitor treatment that are in proximity of EGFR and potentially interact with EGFR in the extracellular space. This is a potentially interesting method and approach to screen for novel extracellular interacting complexes. While my expertise is not sufficient to comment on technical details of the robustness of the methodology, I have several conceptual concerns related also to the physiological relevance of some of the potential EGFR interactors.

Response: We thank you for the considerate comments and important concerns regarding the novelty of this study.

1) Have the authors shown that the EGFR-FabID/EGFR-mAbID do not internalize after binding to EGFR?

Response: We thank the reviewer for the comment on concerns regarding the cellular uptake of EGFR-FabID/EGFR-mAbID after binding to the EGFR. We think it is very important to show where EGFR-FabID is performing the biotinylation reaction, as the reviewer suggested. To address this concern, cells were stained with Streptavidin-Alexa488 without permeabilization after biotinylation by EGFR-FabID. The staining cells were used for FACS analysis to confirm the increase in fluorescently labelled cell populations. FACS results showed an increased cell population with cell surfaces biotinylated by EGFR-FabID (Supplementary Fig. 2a). This indicates that EGFR-FabID biotinylates proteins on the cell membrane after binding to EGFR. Biotinylation on the cell surface has also been confirmed by immunostaining in Fig. 4b. From these results, we conclude that EGFR-FabID performs

biotinylation on the plasma membrane. The revised manuscript discusses the results on FACS. (Revised Supplementary Fig.2) (lines 195-203)

2) With the Expi293F cell line grown in suspension and transiently overexpressing EGFR the authors find that also intracellular sites of the EGFR are biotinylated. How can this happen? Is this due to internalization of the complex when bound to EGFR? The explanation provided on page 16 lines 284-285 seems a bit weak....

Response: We appreciate the reviewer for the comment on concerns regarding the biotinylation of the intracellular domain of EGFR in suspension Expi293F cells transiently overexpressing EGFR. We were as surprised as the reviewers about the biotinylation of the intracellular domain. To address this question, we analysed EGFR-FabID-induced biotinylation by mass spectrometry in Expi293F cells stably expressing EGFR. Although the intracellular part of the EGFR was biotinylated in Expi293F cells transiently overexpressing EGFR, all biotinylation sites on EGFR were extracellular region in Expi293F stably expressing EGFR. Transient overexpression may cause cell injury-mediated EGFR leakage or unusual membrane insertion of EGFR. In the revised manuscript, we added mass spectrometry results of Expi293F cells stably expressing EGFR. We also added a sentence indicating that the biotinylation of the intracellular domain of EGFR may be caused by transient overexpression. (Revised Supplementary Fig. 3) (lines 312-321)

3) Related to 2), why are intracellular sites not biotinylated in cells growing in adherence?

Response: We thank the reviewer for the comment related to 2). While adherent cells and suspension Expi293F cells stably expressing EGFR showed the almost same results. The intracellular biotinylations by EGFR-FabID were only observed in suspension Expi293F cells transiently overexpressing EGFR. This could be due to the transient overexpression of EGFR as described above.

4) In all the biotinylation experiments with Gefitinib, the control with Gefitinib alone is missing. It would be important to show whether extracellular EGFR interactions are affected in catalytically inactive EGFR in the absence of ligand.

Response: We thank you for your constructive comments on the biotinylation experiments. We agree with you and incorporated this suggestion in our paper. The exPPI analysis of EGFR by EGFR-FabID was performed in A431 and NCI-H226 cells treated with Gefitinib alone. EGFR was biotinylated by EGFR-FabID and exPPIs could be detected in the Gefitinib alone treated cells. In the revised manuscript, we added the mass spectrometry

results of A431 and NCI-H226 cells treated with Gefitinib alone. (Revised Supplementary Fig. 9) (lines 412-425)

5) Among the interactors found there is EEF1A1, a translation elongation factor and an intracellular protein. It is extremely surprising that it is found outside the cell in the extracellular space and to interact with EGFR.... Is EEF1A1 leaked out from dying cells or how do the authors explain these results? Is this a specific and relevant interaction and what is the physiological function should this have?

Response: We thank you for the comment regarding the finding of EEF1A1 as a proximal protein of EGFR. We were also surprised that the translation elongation factor EEF1A1 was found as a proximity exPPI protein of the EGFR. In response to the reviewer's comments, we performed an MTS assay to investigate the cytotoxicity of EGFR-FabID. The results showed that cell death was not induced by the addition of EGFR-FabID. EEF1A1 was reported to be present on the plasma membrane and has been shown to play a role in fibronectin-mediated cell regulation (Itagaki et al., J Biol Chem 287(19):16037-46). These findings suggest that EGFR-FabID would biotinylate EEF1A1 on the plasma membrane. In the revised manuscript, we described the results of the MTS assay and the presence of EEF1A1 on the plasma membrane. (Revised Supplementary Fig. 2) (lines 200-203, 352-358)

6) All the results shown suggest interactions with EGFR which are however not demonstrated to occur in live cells. Moreover, proximity is no proof of interaction. Without a direct biological proof, the results could simply be an artifact. Some interactions with truncated proteins (extracellular portion) are shown in Fig. 6c, but to exclude artifacts, experiments in a more physiological setting should be performed, e.g. IF, live imaging, etc., demonstrating also the physiological relevance.

Response: We thank you for constructive comments regarding EGFR interacting proteins. As the reviewer pointed out, it is very important to confirm whether proteins biotinylated by EGFR-FabID interact with EGFR under physiological conditions. However, techniques for analysing direct interactions between membrane proteins under physiological conditions are lacking. We confirmed this using the NanoBiT method, which allows protein interaction analysis on membranes of live cells (Fig. 6d). As the NanoBiT method required overexpression of membrane proteins, additional experiments were performed with the immunofluorescence that could confirm the interaction in a more physiological environment. We performed the immunostaining of INSR, ADAM17, and PTK7 to confirm their colocalization with endogenous EGFR in cells. ADAM17 and PTK7 were colocalized with EGFR in A431 cells (Supplementary Fig. 10a). INSR was also colocalized with EGFR

in NCI-H226 cells (Supplementary Fig. 10b). These biochemical and cell biological interaction analyses suggest that INSR, ADAM17, and PTK7 interact with EGFR. In the revised manuscript, we added the results and descriptions of the immunostaining. (Revised Supplementary Fig. 9) (lines 441-448)

7) It is surprising that EGFR ligands themselves were not biotinylated in these assays. These would have been the perfect positive control for the approach.

Response: We thank the reviewer for the careful review of the manuscript. As the reviewer pointed out, we were also surprised that EGF was not biotinylated by EGFR-FabID. This may be partly because the EGF ligand has only two lysine residues, but a more detailed discussion is needed. To understand the reason why EGF was not biotinylated by EGFR-FabID, a model illustration of EGFR-FabID bound to the EGFR-EGF complex was generated using the crystal structure and AlphaFold (Supplementary Fig. 5). The proximity biotinylation enzyme biotinylates accessible lysines that are exposed on the protein surface because it releases a short-lived, hydrophilic biotinoyl-5'-AMP. EGF ligand contains lysine residues at positions 28 and 48. The structural model suggests that K48, which is located inside EGFR, is not easily biotinylated. K28 is also located at the boundary between the light chain of EGFR-FabID and EGFR, which is also thought to be difficult for hydrophilic biotinoyl-5'-AMP to access. These results suggest that no biotinylation of EGF by EGFR-FabID is due to the lack of exposed lysine residues on the ligand surface. In addition, some biotinylated peptides are not detectable by mass spectrometry due to the degree of ionization. For these reasons, we consider that biotinylated EGF by EGFR-FabID was not detected. In the revised manuscript, we added sentences in the text regarding the lack of biotinylation of EGF. We also added a Supplementary Fig 5. (Revised Supplementary Fig. 5) (lines 380-391)

Minor:

- Was the experiment shown in Fig 3 with Expi293F cells done in the presence or absence of EGF? This is not clear from the text and in Fig3a the scheme contains also EGF.

Response: We thank you for your comment regarding the experiment with Expi293F cells. In the experiment with Expi293F cells, we used serum-free medium without EGF. Therefore, we removed EGF from Fig. 3a. In the revised manuscript, we also specified that serum-free medium is used.

- The author could indicate the sites recognized by EGF making it more evident that EGF and EGFR-FabID do not interfere when given together.

Response: We appreciate you for these constructive comments regarding the EGF recognition sites. As the reviewer suggested, we also thought it was very important to show the site where EGF recognises EGFR, so we added the information. It is known that three EGFR sites (site1,2,3) are involved in the recognition of EGF (Ogiso cell 2002). EGFR-FabID is an antibody that recognizes amino acids 377 to 386 located at site 2. We examined whether EGFR binding of EGFR-FabID affects EGF-EGFR complex formation. In experiments in which EGFR-FabID was added after the addition of EGF, autophosphorylation of EGFR was not decreased. This indicated that EGFR-FabID did not interfere with EGF binding to EGFR. It has also been shown that the large β -sheets at sites 1 and 3 of EGFR are important for the interaction between EGFR and the EGFR ligand TGF- α , which has a similar structure to EGF (Thomas Garrett 2002 cell). The fact that binding of EGFR-FabID to site 2 did not decrease EGFR phosphorylation suggests that the EGF-EGFR complex is stable at the binding of EGFR site 1 and site 3 to EGF. In the revised manuscript, the description of the sites recognized by EGF has been added in the text (lines 277-281, 286-290).

- the English should be improved

We thank you for your comment. The English was corrected for spelling and wording errors, as pointed out by the reviewers. The manuscript was also edited by Editage (www.editage.com), an English editing company.

Reviewer #2 (Remarks to the Author):

The manuscript by Yamada et al. reports the develop a method to uncover extracellular protein-protein interactions. For this they use the AirID proximity labeling enzyme their group previously developed. Their approach fuses AirID to either full length IgG antibody (mAbID) or fragments of antigen binding (FabID). FabID outperformed mAbID in tests using anti-AGIA mAb or EGFR mAb. They go on to nicely show that EGFR-FabID can identify extracellular interactors of EGFR. interestingly, they also show that this approach can identify proteins that interact with EGFR dependent on treatment with EGF ligand or and EGFR inhibitor (gefitinib).

Response: We thank the reviewer for their kind words about our manuscript.

- Overall, the manuscript is nicely written and technically sound. Though the use of AirID, mAbID, and FabID can get a bit confusing.

Response: We thank the reviewer for this constructive comment on the manuscripts. In response to the comments, the revised text in the introduction has been corrected to show the difference between AirID, FabID and mAbID. (lines 91-94)

- This approach relies on having well characterized monoclonal antibodies for the protein of interest, which limits how widely it can be adopted. It is mentioned in the introduction that previous BioID based methods to identify extracellular interactors used genetic insertions. Perhaps the authors can expand on how the monoclonal antibody approach is needed or improves direct fusions to the protein of interest.

Response: We appreciate these helpful suggestions. BioID-based exPPI analysis has been performed by expressing proteins with BioID genetically incorporated between the domains of the extracellular regions of membrane proteins [Shafraz, O. et al., Proc Natl Acad Sci USA. (2020)]. Although this approach has revealed important exPPI results, it has several drawbacks. Many membrane proteins have a signal peptide at their N-terminus that is required for membrane-spanning and prevents insertion of the BioID enzyme to the N-terminus. This means that exPPIs of membrane proteins with N-terminus exposed to the extracellular space must integrate BioID between the domains of the extracellular region. However, the integration of BioID enzymes between the domains of membrane proteins is likely to affect the exPPI of membrane proteins. In addition, BioID-fused proteins do not

allow exPPI analysis of endogenous membrane proteins in their native forms. In this regard, FabID can analyse exPPI of endogenous membrane proteins, although it requires well-characterised antibodies for the target protein. Indeed, EGFR-FabID was able to analyse exPPI of endogenous EGFR (Fig. 5). The advantage of the FabID method is that it can analyse exPPI of endogenous membrane proteins. In the revised manuscript, this content has been added to the Introduction and Discussion sections. (lines 85-89 and 488-503)

- The authors find a number of enriched EGFR interactors it would be nice to show if any are biologically meaningful, but perhaps this is beyond the scope of this manuscript.

Response: We thank you for your comment. This is an important point. We performed immunostaining of INSR, ADAM17, and PTK7 to confirm their colocalization with endogenous EGFR in the cells. ADAM17 and PTK7 were colocalized with EGFR in A431 cells (Supplementary Fig. 9a). INSR was also colocalized with EGFR in NCI-H226 cells (Supplementary Fig. 9b). Biochemical analysis of the NanoBiT method and AlphaScreen and cell biological interaction analysis by immunostaining suggested that INSR, ADAM17 and PTK7 interact with EGFR. PTK7 and INSR share downstream signalling with EGFR [Cui, N.P. et al., *Front Oncol.* 11, 699889 (2021)., Stefani, C. et al., *Int J Mol Sci.* 22, 10260 (2021)., Wang, Y.P. et al., *Cancer Lett.* 337, 96-106 (2013).]. These downstream signalling may be related to EGFR interactions. However, as the reviewer pointed out, we consider that proof of this content is beyond the scope of this manuscript. In the revised manuscript, we added the immunostaining results. (Revised Supplementary Fig. 9)

Reviewer #3 (Remarks to the Author):

Proximity extracellular protein-1 protein interaction analysis of EGFR using AirID-conjugated fragment of antigen binding

Overall Comments:

This manuscript describes a novel and sensitive approach to detect proximity between the extracellular domains of proteins from both adherent and suspension cell lines. The epidermal growth factor receptor was used to develop the method, and the results were validated in multiple cell lines and several key interactions that were previously unknown were confirmed by orthogonal detection methods. By linking biotinylation events to specific peptides, the mass-based readout facilitates structural interpretations. Additionally, the method was demonstrated to quantify changes in multiprotein complexes that result from treatments with a ligand and small molecule. Thus, this manuscript introduces a valuable new method to the biomedical research community that could be applied to interrogate many complex physiological systems. Overall this manuscript is mostly well-written and clear to understand; however, I would recommend that the authors carefully review the text for errors and make additional edits to the grammar to improve the clarity. In many places throughout the manuscript and figure legends it is stated “three biological replicates” but it is not clear what is being shown in the figures. Are the data shown (1) from a single experiment that is “representative” of three biological replicates or (2) the mean of three biological replicates? Please clarify this throughout the manuscript and show the mean +/- standard deviation or standard error of the mean (as appropriate) wherever replicate data exists. Also, there are a number of discrepancies between the figures and their legends. Please review these carefully and ensure that the legend corresponds to the figures as shown. Also, please make sure that all acronyms and abbreviations are defined.

Response: We appreciate the reviewer for the careful review of the manuscript.

Specific Comments about the Main Text:

1. Page 7, lines 116 – 117: “Biotinylation of these purified proteins was mainly found in the heavy chain but not in each light chain (Fig. 1c).” Please mark the positions of the light and heavy chains in Figure 1(c).

Response: We thank you for bringing up these constructive comments. Light and heavy chains are marked in Figure 1c.

2. Page 10, line 169, “To understand why FabID was more highly biotinylated on cells than mAbID, ...”. I think the authors meant to state, “To understand why the EGFR is more highly biotinylated on cells by EGFR-FabID than EGFR-mAbID”?

Response: We thank you for the constructive comments. We received feedback and changed the sentence to "To understand why the EGFR is more highly biotinylated on cells by EGFR-FabID than EGFR-mAbID, we performed structural modelling of FabID and mAbID with the extracellular region of EGFR, as previously reported." (lines 176-178)

3. Page 10, lines 176 - 177: “The productivity of TurboID-fusion Fab was lower than AirID-based EGFR-FabID in Expi293F cells (IB: His in Supplementary Fig. 1b), ...” Should the word “productivity” be replaced with “expression” based on the anti-His immunoblot, which shows a greater abundance of EGFR-FabID?

Response: We appreciate the reviewer’s suggestions. We rephrased the text in the revised manuscript. (line 184)

4. Pages 11 – 12, lines 187 – 199: “MS analysis indicated that EGFR-FabID biotinylated five and seven lysine residues in the extracellular and intracellular regions of EGFR, respectively (Fig. 2f)”. It is not clear to me how the EGFR-FabID was able to biotinylate the seven intracellular lysine residues. Is it thought that this occurred after cell lysis? Also, what distances are the residues K716 – K1188 from Ser380? Is it possible that, following cell lysis, biotinylation occurs "in trans" among EGFR proteins and independent of proximity among both intracellular and extracellular regions of proteins? If so, that will impact the interpretation of extracellular peptides that are found to be biotinylated. If my interpretations are valid, I wonder if it is possible or if the authors considered to quench the enzyme prior to cell lysis?

Response: We thank you for the comment regarding the biotinylation of the intracellular domain of EGFR in Expi293F cells transiently overexpressing EGFR. The issues you raise are very important. To address this question, we analyzed biotinylation by EGFR-FabID by mass spectrometry in Expi293F cells stably expressing EGFR. The results showed that all EGFR biotinylation sites were in the extracellular region. Biotinylation and cell lysis of

Expi293F cells stably expressing EGFR were performed under the same conditions as EGFR-overexpressing cells. It is unlikely that biotinylation occurred after cell lysis because cells were lysed in 6 M guanidine-containing buffer. Therefore, the biotinylation of the intracellular domain of EGFR may be due to damage to the plasma membrane caused by transfection of the plasmid or unusual membrane insertion of overexpressed EGFR. Sentence regarding the biotinylation of the intracellular domain of EGFR by transient overexpression was also added. (Revised Supplementary Fig. 3) (lines 312-318)

5. Page 13, line 222: Why are the 10 new proteins being referred to as “clones”?

Response: We thank you for the careful review. We rephrased the word to “proteins” in the revised manuscript. (line 240)

6. Page 13, lines 227 – 228: I am not sure what is meant by “mediated by one another protein”. Please clarify this statement.

Response: We thank you for the careful review. We corrected the text in the revised manuscript (line 245-248).

7. Page 13, line 229: “...proximal exPPIs induced by EGFR-FabID...” I don’t think that EGFR-FabID is supposed to be inducing interactions, it is providing a way to detect them, right?

Response: We appreciate you for the careful review. We corrected the text in the revised manuscript (lines 245-248).

8. Page 21, line 377: Proximity is misspelled as “ploximity”.

Response: We thank you for the careful review. We corrected the misspelling in our revised manuscript. (line 451)

9. Page 21, lines 377 – 378: “As shown in Fig. 6b, proximity exPPI of PTK7 with EGFR was reduced by gefitinib treatment.” In Figure 6b based on the data from A431 cells, it looks like biotinylation (i.e. proximity) to PTK7 is increased with gefitinib treatment, not reduced; however, it appears that biotinylation (i.e. proximity) to PTK7 is reduced with gefitinib

treatment in NCI-H226 cells. Please clarify this statement in the text. Additionally, it is not clear in Figure 6 (e) or the figure legend that the data are from A431 cells.

Response: We appreciate you for your important comments on the biotinylation changes of PTK7 by the addition of gefitinib. We clearly mentioned in the manuscript and Figure 6e that the analysis was performed using A431 cells. We were also concerned about the discrepancy between the PLA signal value of PTK7 and the biotinylation level. Unfortunately, after our careful analysis, we found that the PLA signal of PTK7 showed a very high background even before treatment with ligand or drug. We think this high background is a major limitation of using the PLA method to detect small changes in interactions caused by the presence of drugs or ligands. Fortunately, the background of ADAM17 and INSR was not as high as that of PTK7, so ADAM17 and INSR roughly matched the PLA signal and biotinylation changes. In the revision, we added a sentence to the Result explaining these things and added a notation in Fig. 6e indicating that A431 was used. In the revision, we added these explanations within the Results section (lines 449-458) and in Figure 6e that A431 was used for the experiment (Fig.6e).

Specific Comments about the Materials and Methods Section:

10. Please comment about how all of the cell lines were authenticated.

Response: We thank the reviewer for the comment on cell stocks. RRID information has been added for all cell lines.

11. All reagents, supplies, kits, and materials listed must include the manufacturer and catalog number, as well as lot numbers wherever possible. In addition, Research Resource Identifiers (RRIDs) (<https://scicrunch.org/resources>) must also be included wherever possible to clarify exactly which materials, supplies, and reagents were used. The RRIDs are particularly important to specify the exact antibodies and cell lines used. All of this information is critical to enable others to leverage this valuable new proximity method being described and to improve the reproducibility of biomedical research in general.

Response: We thank you for the careful review. As pointed out by the reviewer, the inclusion of the RRID/catalogue number is important to make this new proximity method available to others. In the revised manuscript, the RRIDs are listed for the antibodies used and catalogue numbers for all reagents used.

12. Page 28, line 556: Each is misspelled as “eath”.

Response: We appreciate the reviewer for the careful review. We corrected typographical errors in our revised manuscript. (line 694)

13. Page 34, lines 700 – 703: Only volumes are given for the two types of AlphaLISA beads, but the final concentrations are what is important. Also the filters used for this assay in the Envision microplate reader should be described. Finally, it might be more appropriate to refer to a “chemiluminescent signal” than a “luminescence signal”.

Response: We thank the reviewer for your comments on AlphaScreen. In the revised manuscript, we included the final concentration regarding AlphaScreen beads. We also added a description of the filter. We changed the term chemiluminescence signal as it is more appropriate to refer to it as chemiluminescence signal, as the reviewer pointed out. (lines 845-853)

14. Page 35, lines 712 – 713: Please indicate the BZ-X810 microscope settings, including excitation/emission filters and excitation wavelength(s).

Response: We thank the reviewer for the comment regarding microscope filters. We added the filter information to the revised manuscript. (lines 888-892, 922-925)

Specific Comments about the Figures:

15. Figure 1. In panel (c) detection of AGIA-FabID and AGIA-mAbID by anti-polyhistidine is not mentioned in the figure legend. What is FG-DRD1-C? Is it FLAG-GST-DRD1? Please clarify this in the legend. In panel (c) please mark the positions of the light and heavy chains. In panel (h), please show the position of FLAG-p53 in the Western IB: Biotin blot.

Response: We thank the reviewer for the careful review of Fig. 1. We added a sentence regarding the detection of AGIA-FabID and AGIA-mAbID by anti-polyhistidine in the figure legend as the reviewer suggested. We also changed the description of FG-DRD1-C in Fig. 1f to FLAG-GST-DRD1 CTD and renamed Fig. 1e to FLAG-GST-DRD1 CTD also. A description of the FLAG-GST-DRD1 CTD has also been added to the revised manuscript. (Revised Fig.1e) (lines 124-127, 1144-1146)

16. Figure 3. In panel (a) should the 10 new molecules identified be referred to as “proteins” rather than “clones”? I do not see panel (e) in the figure provided. I also do not see any data for Alphascreens with the EGFR extracellular domain and various FLAG fusion proteins. It appears that (d) shows the pathway analysis of extracellular proteins detected using mass spectrometry. Please revise this figure and associated legend.

Response: We thank the reviewer for the careful review of Fig. 3. In panel (a), we replaced clones with proteins. In addition, the legend of the alpha screen data using the EGFR extracellular domain and various FLAG fusion proteins was incorrect and has been deleted. The figure and associated legend in panel (d) were revised. (Revised Fig. 3)

17. Figure 4. In panel (c) data is shown for A431, but not NCI-H226 cells, whereas the legend indicates data are shown for both. In panel (d) “transiently” is misspelled. The data in panel (e) should not be shown as a heatmap – the data should be graphed as a scatterplot with all replicate data points shown.

Response: We appreciate the reviewer for the careful review of Fig. 4. In the revised text, the legend in panel (c) has been revised. We corrected a spelling error in panel (d). A scatterplot of panel (e) was produced and added to Fig. 4e. (Revised Fig. 4)

18. Figure 5. In panel (b) “transiently” is misspelled (in green).

Response: We thank the reviewer for the careful review of the Figure. In the revised figure, we corrected the spelling error in panel (b). (Revised Fig. 5)

19. Figure 6. In panel (b) this data should not be shown as a heatmap – the data should be graphed as a scatterplot with the three replicate data points shown with error bars (as in panels c - e). Doing this might require the data for A431 and NCI-H226 to be shown in separate graphs/panels. In panels c – e, the data show replicates and error bars. Do the error bars indicate standard deviation or standard error of the mean? Please define “PLA” as “proximity ligation assay”. Finally it is not clear in panel (e) or the figure legend that the data are from A431 cells. Please clarify this in both the figure and in the legend.

Response: We thank the reviewer for the careful review of the Figure. Regarding panel (b), we produced a scatter plot and added it to Supplementary data 7. I also added the mean values to the heatmap. We noted in the legend that the error bars in panels c~e represent standard deviations. We defined PLA as proximity ligation assay in the legend and in the manuscript. In panel (e), it is clarified in the figure and in the legend that the data were obtained from A431 cells. (Revised Fig. 6e) (Revised Supplementary Fig.7) (lines 449-450, 1233-1234)

(lines 449-450: Next, the change in proximity exPPIs in EGF, with or without gefitinib treatment, was validated by proximity ligation assay (PLA) method in A431 cells.

lines 1233-1234: Proximity ligation assay (PLA) performed on A431 cells.)

20. Supplementary Fig. 2: Were the data shown in Tables a – c (by the way, “c” is not labeled) from single experiments or were replicates performed? Please clarify that in the figure legend. If replicates were performed, please indicate the type (independent or technical replicates) and include all the data (mean +/- standard deviation) within the tables.

Response: We thank the reviewer for the comment on Supplementary Fig. 2 (Supplementary Fig. 3 in the revised manuscript). In all mass spectrometry analyses, biotinylation is performed by EGFR-FabID in three or five independent reaction systems. In the revised manuscript, the figure legend clearly states that independent replicates by EGFR-FabID and technical replicates by mass spectrometry are performed. Mean values and standard deviations are also shown in the figure. (Revised Supplementary Fig.3)

Reviewer #1 (Remarks to the Author):

The authors have made an effort to address the concerns raised, which have clearly improved the manuscript and their conclusion. However, I am still a bit concerned about the interactions found and possible related artifacts. Although investigating the biological relevance of the interactions found might go beyond the scope of the current paper, I still think this is very important to show, as it provides the necessary validity to the approach, which might in future be used by other groups to identify important interactions.

Please find below my comments/concerns to the responses provided by the authors:

Ad 1) This FACS experiment should demonstrate that EGFR-FabID/EGFR-mabID is not internalized. However, it is not clear from the explanation provided how exactly the experiment was done. And it is also not clear to me how membrane and/or intracellular staining can be distinguished by FACS. The authors mention that they use Streptavidin-Alexa 488 on non-permeabilized cells to reveal where biotinylation takes place. And the staining seems to occur on the cell surface of cells labelled with EGFR-FabID when Biotin is added, which is fine and what one would expect. However, this still does not demonstrate that the complex is also not internalized. The majority will be for sure on the cell membrane (which is what is shown by FACS), but the internalized part might be less and, even if abundant, not detectable by FACS. The staining shown in Fig. 4b (already present in the first submission) is not convincing and this is why I asked for more data that would exclude internalization.

Ad 2/3) it is appreciated that intracellular interactions are not present anymore when cells stably overexpress EGFR, and I still find it very strange that it does so when EGFR is transiently overexpressed. The explanation given, about injured membrane also does not sound very plausible to me.

Ad 5) The authors show an MTS assay to exclude cytotoxicity and want to use it as evidence that EEF1A1 is on the cell membrane, as this has been published by others before. The authors should provide direct evidence for it by staining for EEF1A1 and show that it is on the cell membrane in their system and that it interacts with EGFR. Otherwise, this could all happen intracellularly because the complex is internalized.

Ad 7) Thanks for the explanation provided, which seems plausible, but what about other EGFR ligands, e.g. TGF α , HB-EGF; maybe they expose the required lysines and would be a perfect positive control. This could be checked to additionally support their claims.

Reviewer #2 (Remarks to the Author):

The authors have addressed my comments.

Point-by-Point Responses to the Reviewers' Critiques

Reviewer #1 (Remarks to the Author):

Review Kohdai Yamada et. Al.

The authors have made an effort to address the concerns raised, which have clearly improved the manuscript and their conclusion. However, I am still a bit concerned about the interactions found and possible related artifacts. Although investigating the biological relevance of the interactions found might go beyond the scope of the current paper, I still think this is very important to show, as it provides the necessary validity to the approach, which might in future be used by other groups to identify important interactions.

Response: We thank you for your kind comments and valuable suggestions on our revised manuscript.

Ad 1) This FACS experiment should demonstrate that EGFR-FabID/EGFR-mabID is not internalized. However, it is not clear from the explanation provided how exactly the experiment was done. And it is also not clear to me how membrane and/or intracellular staining can be distinguished by FACS. The authors mention that they use Streptavidin-Alexa 488 on non-permeabilized cells to reveal where biotinylation takes place. And the staining seems to occur on the cell surface of cells labelled with EGFR-FabID when Biotin is added, which is fine and what one would expect. However, this still does not demonstrate that the complex is also not internalized. The majority will be for sure on the cell membrane (which is what is shown by FACS), but the internalized part might be less and, even if abundant, not detectable by FACS. The staining shown in Fig. 4b (already present in the first submission) is not convincing and this is why I asked for more data that would exclude internalization.

Response: We thank the reviewer for the comment on internalization. First, regarding the FACS experiment shown in the previous revision, we described in the Methods section again what kind of experiment was performed (lines 761-769). As the Reviewer mentioned, EGFR internalization cannot be completely identified by FACS. However, the FACS results show that the cell surface is stained with Streptavidin-Alexa488, indicating that the cell surface is

biotinylated. If EGFR-FabID biotinylation is dependent on EGFR endocytosis, then endocytosis inhibitors would disable EGFR biotinylation by EGFR-FabID. We conducted experiments with a well-known endocytosis inhibitor Dynasore to exclude the possibility that endocytosis of EGFR is important for biotinylation by EGFR-FabID. After the treatment with Dynasore to inhibit endocytosis, NCI-H226 cells were biotinylated with EGFR-FabID. As a result, immunoblotting of biotinylated cell lysates showed that EGFR was biotinylated even in the endocytosis-inhibited cells. This indicates that the EGFR-FabID can biotinylate the proximity proteins in the condition without endocytosis. The FACS results and the additional experiments using Dynasore strongly suggest that endocytosis is not essential for the FabID reaction. Furthermore, the band intensity in the immunoblotting was similar between treatments with and without Dynasore, indicating that the majority of biotinylated EGFR is present at the cell surface even under conditions of endocytosis inhibition. In addition, the complex of EGFR-FabID and EGFR is considered to be quite stable since it is a binding of a high affinity antibody (clone EMab-134, references 34 and 35 in the manuscript) to an antigen. Taken together, this result seems to suggest that the majority of the complex of EGFR-FabID and EGFR is localized at the cell surface. We appreciate the Reviewer's astute insights, which led us to a new way of thinking about the use of EGFR-FabID. In the revised manuscript, we described the results biotinylation with EGFR-FabID during endocytosis inhibition. (Revised Supplementary Fig. 2d) (line 272-280, 784-791)

Ad 2/3) it is appreciated that intracellular interactions are not present anymore when cells stably overexpress EGFR, and I still find it very strange that it does so when EGFR is transiently overexpressed. The explanation given, about injured membrane also does not sound very plausible to me.

Response: We appreciate the reviewer for the comment on concerns regarding the biotinylation of the EGFR intracellular domain by transiently overexpressing EGFR. We agree with the Reviewer that additional experiments are needed to further explore this point. We performed the immunoblotting by separating and collecting cells and culture medium after transfection with EGFR or pcDNA3.1 empty vector (Reviewer Figure A shown below in this section). Transfection was performed under the conditions used for mass spectrometry and culture time was the same. As a result, EGFR was detected in both the cell lysates and

culture medium of the transiently overexpressing EGFR treatment. Tubulin was also detected in the culture medium. These results indicated that EGFR proteins were present in the culture medium when EGFR was transfected transiently. The fact that Tubulin was also detected in the culture medium suggested that some intracellular proteins would leak out of the cells due to the transfection. Both the extracellular and intracellular regions of EGFR proteins in the culture medium are likely to be exposed in the medium, and then EGFR proteins in the culture medium are biotinylated by EGFR-FabID. This result indicates the reason why the intracellular region of transiently expressed EGFR was biotinylated by EGFR-FabID and detected by mass spectrometry shown in Supplementary Fig. 3b. We thank the Reviewer for the deep insights into the biotinylation of intracellular regions in transiently overexpressed EGFR.

Reviewer Figure A, Detection of EGFR and tubulin proteins in the culture medium after transiently overexpressing EGFR.

Transfection was performed using the same protocol as when LC-MS/MS was performed for biotinylated protein analysis using Expi293F cells transiently overexpressing EGFR. The pcDNA3.1 vector was transfected for mock and pcDNA3.1-EGFR for EGFR. Cells and culture medium were collected 24 hours after transfection and were analyzed by the immunoblotting. Sup represents culture medium and cell represents cell lysate.

Ad 5) The authors show an MTS assay to exclude cytotoxicity and want to use it as evidence that EEF1A1 is on the cell membrane, as this has been published by others before. The authors should provide direct evidence for it by staining for EEF1A1 and show that it is on the cell membrane in their system and that it interacts with EGFR. Otherwise, this could all happen intracellularly because the complex is internalized.

Response: We appreciate your important comments regarding the interaction between EEF1A1 and EGFR. We also thought it was very important to validate that the interaction between EEF1A1 and EGFR occurs at the plasma membrane, as the reviewer suggested. Therefore, we used the immunostaining to confirm that EEF1A1 and EGFR interact on the plasma membrane as described by the reviewer. We used non permeabilized cells for antibody staining to confirm that the interaction was occurring on the plasma membrane. As a result of the immunostaining, it was found that EEF1A1 and EGFR interacted on the plasma membrane. From these results, we determined that EEF1A1 was biotinylated to EGFR-FabID at the cell surface due to its co-localization with EGFR. In the revised manuscript, we added a figure of the immunostaining of EEF1A1 and EGFR and a description of the co-localization of EEF1A1 and EGFR. (Supplementary Fig. 10b) (lines 365-368)

Ad 7) Thanks for the explanation provided, which seems plausible, but what about other EGFR ligands, e.g. TGF α , HB-EGF; maybe they expose the required lysines and would be a perfect positive control. This could be checked to additionally support their claims.

Response: We appreciate your comments regarding biotinylation of EGFR ligands. As the reviewer suggested, I think it is important to consider other ligands as well. There are currently six known ligands for EGFR other than EGF. Among them, TGF- α , Epiregulin, and Epigen have only one lysine in their ligand region, which is difficult to label with biotinylation of ligands. Although we also considered HB-EGF, Betacellulin, and Amphilegulin as the remaining ligands, they are difficult to analyze by mass spectrometry after biotinylation because of their glycosylation. Then, we decided to try biotinylation again using EGF known as the most studied ligand for EGFR. However, as we answered in the 1st revised response, EGF has no surface-exposed lysine (Supplementary Fig. 5a), so we have an idea that the fusion of a lysine-rich tag to EGF, such as FLAG-GST (glutathione S-transferase) tag.

To this end, we constructed a vector of FLAG-GST-EGF (pcDNA3.4-FLAG-GST-TEV-His-AGIA-GS linker-EGF) in the form of a TEV-His-AGIA-GS linker inserted between FLAG-GST and EGF. Fortunately, the FLAG-GST-EGF ligand induced tyrosine-kinase activity of EGFR (Supplementary Fig. 5b), indicating the activity as a ligand. Expi293F EGFR-stable expression cells were cultured for two days after transfection with this vector and biotinylated by EGFR-FabID. The culture medium was analyzed by the immunoblotting after GST pull-down by glutathione-sepharose beads, and biotinylation of EGF was detected (Supplementary Fig. 5c). This result showed that EGFR-FabID can biotinylate the ligand if there are sufficient lysine residues on the ligand surface and suggested that no biotinylation of EGF by EGFR-FabID was due to the lack of exposed lysine residues on the ligand surface. This experiment suggests that the FabID system could be used to detect for ligands of membrane proteins if the ligands have lysine residues that can be biotinylated on their surface. We deeply appreciate the reviewer's precise comments that made this clear. In the revised manuscript, we described the results of the FLAG-GST-EGF biotinylated by EGFR-FabID. (lines 401-414, 884-899)

Reviewer #2 (Remarks to the Author):

The authors have addressed my comments.

Response: We thank you for your kind comments on our revised manuscript.